EMBO
Molecular Medicine

# Reversible immortalisation enables genetic correction of human muscle progenitors and engineering of next-generation human artificial chromosomes for Duchenne muscular dystrophy

Sara Benedetti[1,2], Narumi Uno[3,4,†], Hidetoshi Hoshiya[1,†,#], Martina Ragazzi[1,§], Giulia Ferrari[1], Yasuhiro Kazuki[3,4], Louise Anne Moyle[1], Rossana Tonlorenzi[5], Angelo Lombardo[6], Soraya Chaouch[7], Vincent Mouly[7], Marc Moore[8], Linda Popplewell[8], Kanako Kazuki[4], Motonobu Katoh[4], Luigi Naldini[9], George Dickson[8], Graziella Messina[9,‡], Mitsuo Oshimura[4,‡], Giulio Cossu[10,‡,*] & Francesco Saverio Tedesco[1,‡,**]

## Abstract

Transferring large or multiple genes into primary human stem/progenitor cells is challenged by restrictions in vector capacity, and this hurdle limits the success of gene therapy. A paradigm is Duchenne muscular dystrophy (DMD), an incurable disorder caused by mutations in the largest human gene: dystrophin. The combination of large-capacity vectors, such as human artificial chromosomes (HACs), with stem/progenitor cells may overcome this limitation. We previously reported amelioration of the dystrophic phenotype in mice transplanted with murine muscle progenitors containing a HAC with the entire dystrophin locus (DYS-HAC). However, translation of this strategy to human muscle progenitors requires extension of their proliferative potential to withstand clonal cell expansion after HAC transfer. Here, we show that reversible cell immortalisation mediated by lentivirally delivered excisable hTERT and Bmi1 transgenes extended cell proliferation, enabling transfer of a novel DYS-HAC into DMD satellite cell-derived myoblasts and perivascular cell-derived mesoangioblasts. Genetically corrected cells maintained a stable karyotype, did not undergo tumorigenic transformation and retained their migration ability. Cells remained myogenic *in vitro* (spontaneously or upon MyoD induction) and engrafted murine skeletal muscle upon transplantation. Finally, we combined the aforementioned functions into a next-generation HAC capable of delivering reversible immortalisation, complete genetic correction, additional dystrophin expression, inducible differentiation and controllable cell death. This work establishes a novel platform for complex gene transfer into clinically relevant human muscle progenitors for DMD gene therapy.

**Keywords** DMD; gene therapy; human artificial chromosomes; human muscle stem/progenitor cells; immortalisation
**Subject Categories** Genetics, Gene Therapy & Genetic Disease; Musculoskeletal System; Stem Cells

## Introduction

Duchenne muscular dystrophy (DMD) is the most common muscle disorder of childhood and one of the most severe forms of muscular

1 Department of Cell and Developmental Biology, University College London, London, UK
2 Great Ormond Street Institute of Child Health, University College London, London, UK
3 Department of Biomedical Science, Institute of Regenerative Medicine and Biofunction, Tottori University, Yonago, Tottori, Japan
4 Chromosome Engineering Research Center (CERC), Tottori University, Yonago, Tottori, Japan
5 Division of Neuroscience, Institute of Experimental Neurology, San Raffaele Scientific Institute, Milan, Italy
6 San Raffaele Telethon Institute for Gene Therapy (TIGET), San Raffaele Scientific Institute and Vita Salute San Raffaele University, Milan, Italy
7 AIM/AFM Center for Research in Myology, Sorbonne Universités, UPMC Univ. Paris 06, INSERM UMRS974, CNRS FRE3617, Paris, France
8 School of Biological Sciences, Royal Holloway-University of London, Egham, Surrey, UK
9 Department of Biosciences, University of Milan, Milan, Italy
10 Division of Cell Matrix Biology and Regenerative Medicine, University of Manchester, Manchester, UK
*Corresponding author. Tel: +44 161 3062526; E-mail: giulio.cossu@manchester.ac.uk
**Corresponding author. Tel: +44 2031 082383; E-mail: f.s.tedesco@ucl.ac.uk
†These authors contributed equally to this work
‡These authors contributed equally to this work as senior authors
#Present address: Cell and Gene Therapy Catapult, Guy's Hospital, Great Maze Pound, London, UK
§Present address: MolMed S.p.A, Milan, Italy

dystrophy that leads to progressive muscle wasting and premature death (Mercuri & Muntoni, 2013a). DMD is caused by mutations in the X-linked dystrophin gene, which encodes for a protein responsible for sarcolemma integrity (Hoffman *et al*, 1987). Despite extensive pre-clinical work and many novel clinical trials (Benedetti *et al*, 2013; Mercuri & Muntoni, 2013b; Bengtsson *et al*, 2016), currently there are no definitive treatments. One of the main obstacles to the development of an effective gene therapy for DMD is the large size of the dystrophin gene (2.4 Mb), preventing cloning of its full cDNA (14 kb) into conventional gene therapy vectors. This limitation is then amplified by the abundance of degenerating skeletal muscle needing repair or regeneration.

In recent years, a new generation of large cloning capacity gene delivery vectors named human artificial chromosomes (HACs) has been developed (Kazuki & Oshimura, 2011; Kouprina *et al*, 2014; Oshimura *et al*, 2015; Tedesco, 2015). HACs present several advantages for DMD gene therapy, including (i) the ability to carry large DNA sequences such as entire genetic loci, thus resulting in a more physiological expression of genes with complex transcriptional regulation (including dystrophin), (ii) stable episomal maintenance of a single copy gene, avoiding the risk of insertional oncogenesis. In this context, a HAC containing the entire dystrophin locus (DYS-HAC) was engineered for *ex vivo* stem cell gene therapy of DMD (Hoshiya *et al*, 2009).

In the last two decades, several populations of stem/progenitor cells with myogenic potential have been isolated (Tedesco *et al*, 2010, 2017), but so far only satellite cell-derived myoblasts (Perie *et al*, 2014; Skuk & Tremblay, 2014), muscle pericyte-derived mesoangioblasts (Cossu *et al*, 2015) and (to a minor extent) muscle-derived AC133[+] cells (Torrente *et al*, 2007) have undergone clinical experimentation. We have shown that transplantation of murine dystrophic mesoangioblasts corrected with a DYS-HAC ameliorated the phenotype of dystrophic *mdx* mice (Tedesco *et al*, 2011). These results provided the first evidence of safe and efficacious pre-clinical gene replacement therapy with a HAC into an animal model of a genetic disease, paving the way for translating HAC gene transfer to human cells. However, proliferation of human and murine cells is regulated by different pathways, often resulting in a limited lifespan of human somatic cells. Therefore, primary human muscle cells are likely to require an extension of their proliferative potential to withstand selection of corrected cells and subsequent expansion to

clinically relevant numbers after clonal HAC transfer [in the range of $10^9$ cells (Cossu *et al*, 2015)].

Here, we developed a novel genetic correction strategy based upon the use of reversibly immortalising lentiviral vectors to extend the proliferative potential of human myoblasts and mesoangioblasts. We show that DMD muscle progenitor cells immortalised by means of lentiviral vectors expressing the excisable catalytic subunit of human telomerase (hTERT) and the cell cycle regulator Bmi1 (Cudre-Mauroux *et al*, 2003) enable transfer of a novel DYS-HAC. Moreover, the presence of the herpes simplex virus thymidine kinase (HSV-TK) cDNA between loxP sites allows selective drug-induced elimination of target cells that escaped transgene excision by Cre recombination (Salmon *et al*, 2000). DMD, reversibly immortalised, DYS-HAC-corrected progenitors were transplanted in mouse models of acute and chronic muscle injury, where they engrafted regenerating skeletal muscle. Lastly, we combined all relevant gene functions into a single next-generation synthetic HAC capable of delivering reversible immortalisation, complete genetic correction, additional dystrophin expression, inducible differentiation and controllable cell death, generating the largest and possibly most complex gene therapy vector developed to date.

# Results

## Generation of a novel human artificial chromosome containing the entire human dystrophin locus

To facilitate pre-clinical development of the DYS-HAC platform for human myogenic cells, we engineered a novel DYS-HAC devoid of potentially immunogenic gene products such as the enhanced green fluorescent protein (EGFP), blasticidin (Bsd), HSV1-TK (Tk) and hypoxanthine-guanine phosphoribosyltransferase (HPRT). For simplicity, the DYS-HAC previously engineered (Hoshiya *et al*, 2009) and used in our former murine study (Tedesco *et al*, 2011) is here renamed DYS-HAC1, to distinguish it from the newly generated DYS-HAC2. DYS-HAC2 was engineered by homologous recombination-mediated DYS-HAC1 modifications (Fig 1A). In order to remove EGFP, Bsd, HPRT and HSV1-TK genes from DYS-HAC1 and to add the floxable (FRT) neomycin (Neo) gene for selection and lox71 site and 5′ HPRT for further gene insertion, the targeting vector pN (Fig 1A), containing both the Neo gene and the regions for homologous recombination (A:

**Figure 1. Generation of a novel HAC containing the entire human dystrophin locus by homologous recombination.**

A    The scheme shows a linearised map of the vectors and the strategy used to generate DYS-HAC2 by homologous recombination of DYS-HAC1 (Hoshiya *et al*, 2009). pN targeting vector, which contains regions for homologous recombination (A: 3.8 kb and B: 2.6 kb, in green) and a floxable (FRT) neomycin (Neo), was used to remove extra genes (EGFP, Bsd, HPRT and Tk) on DYS-HAC1 and to insert a floxable Neo gene. Primers designed to amplify DYS-HAC1 or DYS-HAC2 specific regions are highlighted in red.

B    Phase contrast (left) and fluorescence (EGFP, right) images of DT40 cells containing DYS-HAC1 and DYS-HAC2. Scale bar: 50 μm.

C    PCR analyses to discriminate between DYS-HAC1 and DYS-HAC2. DT40 cells: negative control.

D    PCR panel to detect dystrophin exons in DT40(DYS-HAC1) and DT40(DYS-HAC2) cells. DT40 cells: negative control; human mesoangioblasts: positive control.

E    *In situ* fluorescence hybridisation (FISH) analysis of DT40(DYS-HAC2) cells. White arrowheads: DYS-HAC2. Red: rhodamine-human COT-1 DNA; green: dystrophin FITC-DMD-BAC RP11-954B16; yellow: merge. Scale bar: 5 μm. DT40(DYS-HAC2) hybrid was used to transfer the DYS-HAC2 in CHO cells (complete list in Appendix Table S1).

F    FISH analyses of CHO(DYS-HAC2)-7 (left) and A9(DYS-HAC2)-9 (right) clones. White arrowheads: DYS-HAC2. CHO(DYS-HAC2) hybrid was used to transfer DYS-HAC2 in A9 cells (complete list in Appendix Table S2). Red/purple: rhodamine-human COT-1 DNA; green: dystrophin FITC-DMD-BAC RP11-954B16; yellow: merge. Scale bar: 5 μm.

Source data are available online for this figure.

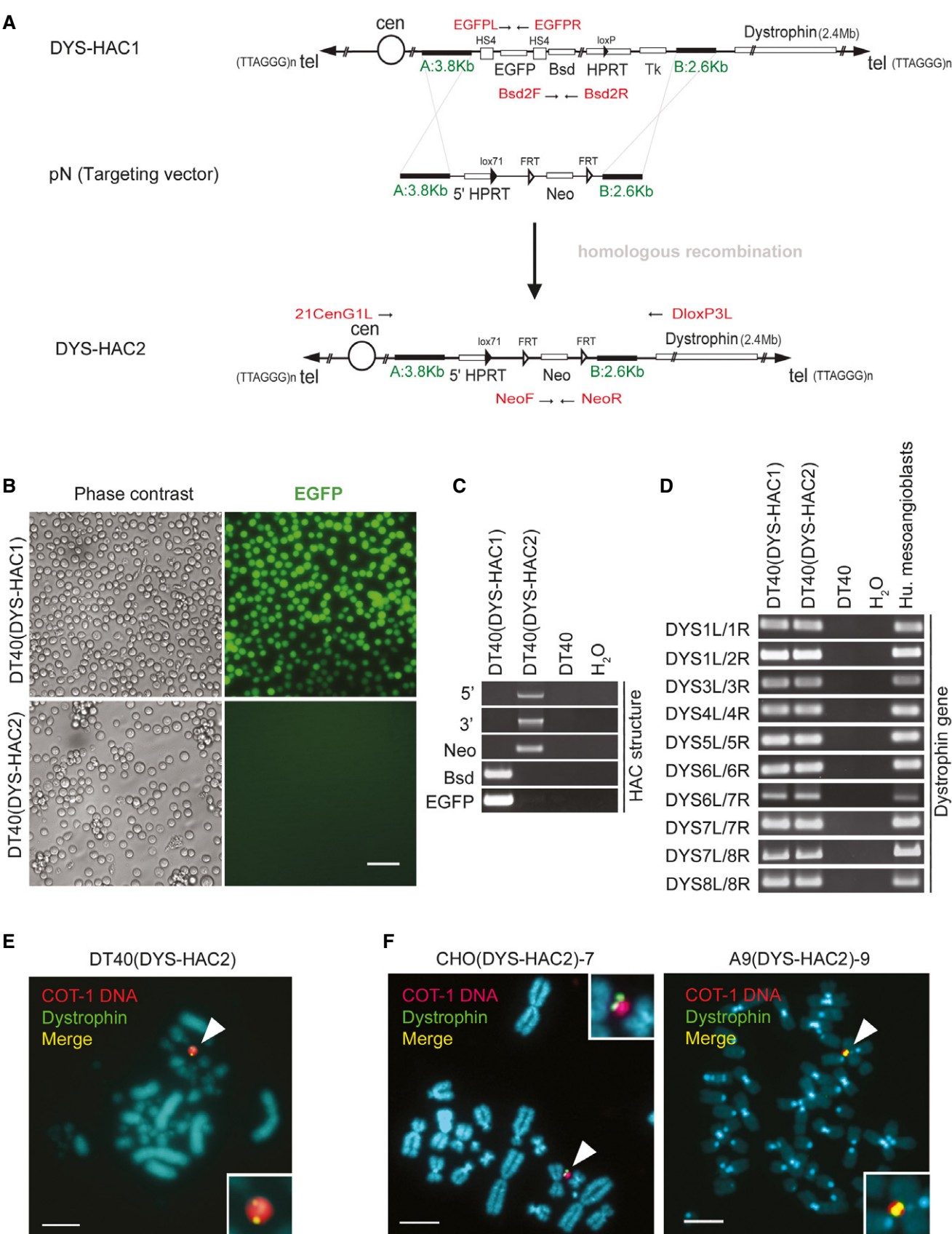

Figure 1.

3.8 kb and B: 2.6 kb), was introduced by electroporation into chicken lymphoid DT40 cells already containing DYS-HAC1 [DT40(DYS-HAC1)]. DT40 hybrids were then subjected to G418 (neomycin) selection, and 19 G418-resistant DT40 hybrids were randomly selected for further analysis. One clone was found to be EGFP negative (Fig 1B), indicating successful homologous recombination. PCRs using primers able to discriminate DYS-HAC2 from DYS-HAC1 confirmed correct targeting (Fig 1C), whereas another set of primers designed on the dystrophin gene confirmed sequence integrity (Fig 1D). Finally, FISH analysis established that DYS-HAC2 segregated independently without host genome insertions or translocations (Fig 1E).

The novel DYS-HAC2 was then transferred via microcell-mediated chromosome transfer (MMCT) from DT40(DYS-HAC2) cells into Chinese hamster ovary (CHO) cells, to scale-up production of microcells for HAC transfer. Twenty CHO(DYS-HAC2) clones were randomly picked after G418 selection and both PCR and FISH confirmed the presence of a single episomal copy of DYS-HAC2 in four clones (Appendix Table S1). Among them, clone CHO(DYS-HAC2)-7 was then selected for transfer of DYS-HAC2 into murine A9 cells, as they have an even higher efficiency in generating microcells compared to CHO cells. Following G418 selection, 27 A9(DYS-HAC2) clones were randomly picked and presence of DYS-HAC2 was confirmed by PCR in five clones. FISH analysis showed episomal presence of DYS-HAC2 in three out of five A9(DYS-HAC2) clones (Appendix Table S2). Figure 1F shows fluorescent *in situ* hybridisation (FISH) images of CHO(DYS-HAC2)-7 and A9(DYS-HAC2)-9 clones utilised as DYS-HAC2 donors in subsequent experiments.

## Reversible immortalisation of DMD myoblasts enables DYS-HAC transfer and complete genetic correction

Combined expression of hTERT and Bmi1 was shown to immortalise human myoblasts (Cudre-Mauroux *et al*, 2003). To test the hypothesis that reversible immortalisation could extend cell proliferation enough to allow HAC transfer in human myogenic progenitors, we planned to transfer the newly generated DYS-HAC2 (Fig 1) into reversibly immortalised DMD myoblasts (riDMD myoblasts). Before proceeding with HAC transfer, we confirmed that riDMD myoblasts (i) contained and transcribed hTERT and Bmi1 transgenes (Fig EV1A), (ii) had maintained their myogenic potential (Fig EV1B), (iii) generated dystrophin-deficient myotubes *in vitro* (Fig EV1C), (iv) were not tumorigenic (*N* = 3; Table EV1). We then transferred DYS-HAC2 into riDMD myoblasts via MMCT. Four G418-resistant clones, namely riDMD(DYS-HAC2)#α, #β, #γ and #δ, were selected. PCR analysis showed that two out of the four clones were positive for all analysed HAC regions (clones #α and #δ; Fig 2A, red boxes). Parental riDMD myoblasts have a deletion from exon 5 to exon 7 in the dystrophin gene (Fig 2B, first lane), which would result into an out-of-frame mutation: this was indeed confirmed by dystrophin transcript analysis, which demonstrated an out-of-frame mutation in riDMD myoblasts and ruled out a potential restoration of the reading frame by skipping of exon 8 (Fig EV1D; Muntoni *et al*, 1994; Cudre-Mauroux *et al*, 2003). Importantly, PCRs for dystrophin exons 5, 6 and 7 after HAC transfer demonstrated that both riDMD(DYS-HAC2)#α and #δ were positive for the exons originally deleted in the parental DMD myoblasts (Fig 2B), showing correction of the dystrophin gene sequence. FISH and karyotype analyses confirmed the presence of a single copy of DYS-HAC2 and

a normal karyotype (Fig 2C). Subcutaneous injection of DYS-HAC2-corrected DMD myoblasts into immunodeficient *scid/beige* mice (*N* = 5 per clone) did not result in tumour formation (*N* = 10; Table EV1). Finally, riDMD(DYS-HAC2)#α myoblasts showed restoration of dystrophin mRNA expression and protein production in skeletal myotubes upon *in vitro* differentiation (Fig 2D–F; detailed analysis of myogenic differentiation in Appendix Fig S1A).

## hTERT and Bmi1 expression prevents replicative senescence of human mesoangioblasts

The experimental work described so far provides proof-of principle evidence of HAC transfer into human dystrophic myoblasts. However, although myoblast transplantation appears to be a promising therapeutic strategy for localised forms of muscular dystrophy (Perie *et al*, 2014), it appears of limited value for widespread muscle disorders such as DMD, where their modest migration potential is a major hurdle (Tedesco *et al*, 2010; Skuk & Tremblay, 2014). To overcome this limitation, we extended the DYS-HAC platform to human mesoangioblasts, which can be delivered systemically via the arterial circulation and have been recently assessed in a phase I/II clinical trial based upon allogeneic transplantation (Cossu *et al*, 2015). Post-natal human mesoangioblasts are considered to be the *in vitro* progeny of a subset of alkaline phosphatase (ALP)-positive skeletal muscle pericytes (Dellavalle *et al*, 2007).

Firstly, human mesoangioblasts were isolated from muscle biopsies of three different healthy subjects (H#1, H#2 and H#3) according to the standard protocols (Tonlorenzi *et al*, 2007). To exclude myoblast cross-contamination, cells were FACS-purified as ALP-positive/CD56-negative (see Materials and Methods). After a short *in vitro* expansion, H#1, #H2 and H#3 human mesoangioblasts were co-transduced with LOX-TERT-IRESTK and LOX-CWBmi1 lentiviral vectors. As an additional control, cells were transduced with a LOX-GFP-IRESTK (Fig EV2A). Phase contrast microscopy revealed that hTERT + Bmi1 transduced polyclonal populations (Fig 3A, upper row, right images) showed a similar morphology to their control (CTR) counterparts (Fig 3A, upper row, left images). One polyclonal population (hTERT + Bmi1 H#3) was then cloned by limiting dilution and three hTERT + Bmi1 clones were selected for further analysis (namely H#3A, H#3B and H#3C; Fig 3A, lower row). PCR analyses performed on genomic DNA of clonal and polyclonal populations confirmed the presence of hTERT and Bmi1 transgenes (Fig 3B). Transcription of both transgenes was then confirmed by RT–PCR (Fig 3C) and quantitative real-time RT–PCR analyses (Fig 3D).

We previously reported that human mesoangioblasts show decreased telomerase activity with prolonged passages in culture (Dellavalle *et al*, 2007). In order to verify whether exogenous expression of hTERT was sufficient to restore telomerase activity in human mesoangioblasts as in other cell types (Weinrich *et al*, 1997; Bodnar *et al*, 1998), a telomeric repeat amplification protocol (TRAP) assay was performed. TRAP assay showed that all hTERT + Bmi1 transduced clones (H#3A, H#3B and H#3C) had detectable telomerase activity after 15 passages in culture (Fig 3E). Telomerase activity was maintained up to at least 40 passages in culture (Fig 3E), and all hTERT + Bmi1 clones and polyclonal populations produced more Bmi1 protein than their untransduced

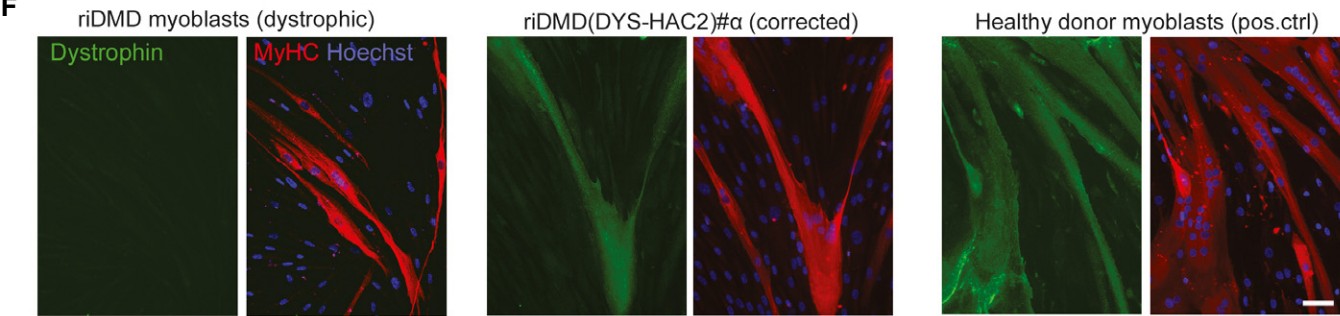

**Figure 2.  DYS-HAC2 transfer into reversibly immortalised DMD myoblasts.**

A   PCR panel for HAC sequences in four neomycin-resistant riDMD(DYS-HAC2) myoblast clones. Red boxes: positive riDMD(DYS-HAC2) myoblasts.

B   PCR analysis of human dystrophin exons on genomic DNA of riDMD myoblasts, riDMD(DYS-HAC2)#α and riDMD(DYS-HAC2)#δ clones. Parental riDMD myoblasts: negative control for exons 5–7; CHO(DYS-HAC2)-7: positive control.

C   Left panel: karyotype analysis of riDMD myoblasts, riDMD(DYS-HAC2)#α and riDMD(DYS-HAC2)#δ myoblast clones. Right panel: FISH analysis on riDMD myoblasts and two selected riDMD(DYS-HAC2) myoblast clones (#α and #δ). Insets: magnifications showing single episomal DYS-HAC2. Red/purple: rhodamine-p11-4 human alpha satellite (centromeres of chromosome 13 and 21, hChr 13/21(cen)); green: dystrophin FITC-DMD-BAC RP11-954B16; yellow: merge. White arrowheads: DYS-HAC2. Scale bar: 3 μm.

D   RT–PCR panel for dystrophin expression in differentiated riDMD(DYS-HAC2)#α. Parental riDMD myoblasts: negative control for dystrophin (exons 5–7); healthy myoblasts: positive control.

E   Western blot for dystrophin (427 kDa) in differentiated riDMD and riDMD(DYS-HAC2)#α myoblasts; human muscle and differentiated human inducible myogenic cells (Maffioletti et al, 2015) were used as positive control; differentiated DMD-inducible myogenic cells (Maffioletti et al, 2015) were used as negative control and myosin heavy chain (MyHC) as normaliser (40 μg of proteins loaded for all samples but human muscle, which had 30 μg loaded).

F   Immunofluorescence images showing in vitro muscle differentiation of riDMD myoblasts (negative control), riDMD(DYS-HAC2)#α and healthy donor myoblasts (positive control). Red: MyHC; green: dystrophin; blue: Hoechst. Scale bar: 50 μm.

Source data are available online for this figure.

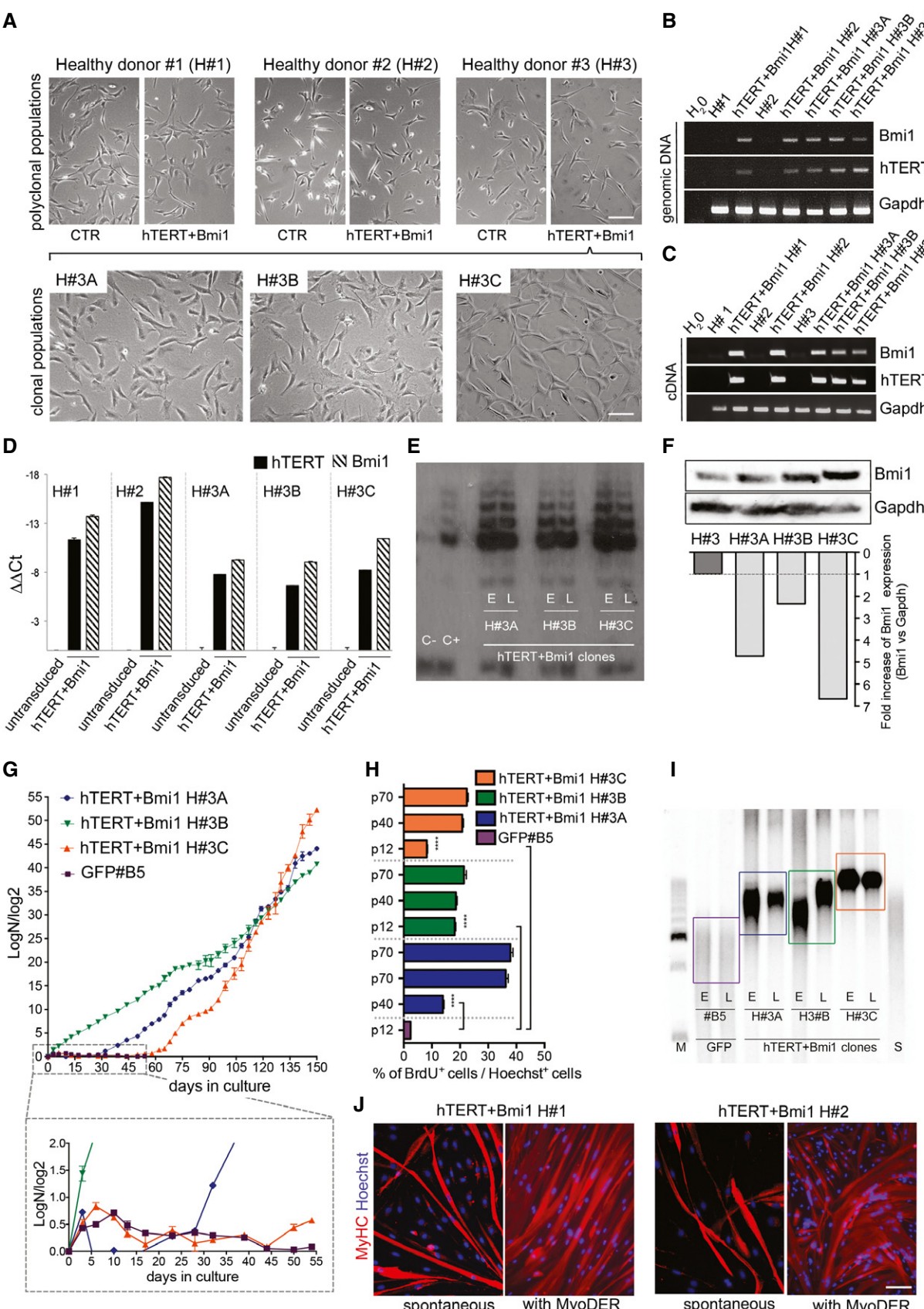

**Figure 3.**

◄

Figure 3.  Bypassing replicative senescence in human mesoangioblasts by lentiviral expression of Bmi1 and hTERT.

A   Upper row: phase contrast morphology of human mesoangioblasts derived from muscle biopsies of three different healthy donors (H#1, #2 and #3) before (CTR, left images) and after lentiviral transduction with LOX-TERT-IRESTK and LOX-CWBmi1 (hTERT + Bmi1, right images). Scale bar: 100 μm. Lower row: phase contrast morphology of three clones selected from hTERT + Bmi1 H#3 polyclonal population: H#3A, H#3B and H#3C. Scale bar: 100 μm.
B   PCR on Bmi1 and hTERT transgenes. Gapdh was used as housekeeping gene.
C   RT–PCR on cDNA for Bmi1 and hTERT. Gapdh was used as housekeeping gene.
D   Quantitative real-time PCR analysis for hTERT and Bmi1 gene expression on untransduced (H#1, H#2 and H#3) and hTERT + Bmi1 polyclonal populations (hTERT + Bmi1 H#1 and H#2) and clones (hTERT + Bmi1 H#3A, H#3B and H#3C). Gapdh expression was used as normaliser. Data are expressed as means ± SEM (n = 2).
E   Telomeric repeat amplification protocol (TRAP) assay performed on hTERT + Bmi1 H#3A, H#3B and H#3C clones at early (E; p17, p14 and p25, respectively, for H#3A, H#3B and H#3C) and late (L, p40) passages. HeLa cells were used as positive control (C+).
F   Bmi1 protein production assessed by Western blot and relative densitometry in a polyclonal parental population (H#3) and hTERT + Bmi1 clones. Gapdh protein used as normaliser.
G   Population doubling curves of hTERT + Bmi1 H#3A, H#3B, H#3C and GFP#B5 clones. The magnification shows the growth curve from day 0 to day 55 for all analysed clones. Data are expressed as means ± SEM (n = 2).
H   Bar graph showing proliferation rate of hTERT + Bmi1 clones (H#3A, H#3B and H#3C) and GFP clone (GFPB#5). Data are expressed as means ± SEM (n = 3). ****P < 0.0001, one-way analysis of variance (ANOVA) with Tukey's post hoc post-test.
I   Telomeric restriction fragment (TRF) assay of hTERT + Bmi1 clones (H#3A, H#3B and H#3C) and GFP#B5 control clone at early (E; p17, p14 and p25, respectively, for H#3A, H#3B and H#3C) and late (L; p40) passages. S: CHQ standard; M: 2.5-kb DNA ladder. Signal for GFP#B5 was weaker as fewer cells were obtained due to reduced proliferation.
J   Immunofluorescence analysis of spontaneous (left) and MyoD-ER-mediated (right) in vitro skeletal muscle differentiation of hTERT + Bmi1 polyclonal populations H#1 and H#2. Red: myosin heavy chain (MyHC); blue: Hoechst. Scale bar: 50 μm.

Source data are available online for this figure.

counterparts (Figs 3F and EV2B). Proliferation analyses demonstrated that hTERT and Bmi1 expression allowed human mesoangioblasts to bypass replicative senescence, whereas untransduced parental cells and GFP control clone #B5 underwent senescence after ~15 passages in culture (Figs 3G and EV2C). Functional BrdU incorporation assay confirmed data obtained in proliferation curves (Figs 3H and EV2D).

Telomerase is responsible for telomere maintenance by adding repeated telomeric sequences to prevent erosion-mediated senescence (Shay & Wright, 2005). Therefore, telomere length was measured in hTERT + Bmi1 clones by means of telomeric restriction fragment (TRF) assay (details in Materials and Methods). As shown in Figs 3I and EV2E, all hTERT + Bmi1 clones had stable and longer telomeres compared to those of the control GFP-transduced clone (GFP#B5).

We then investigated the ability of reversibly immortalised mesoangioblasts to differentiate into multinucleated skeletal myotubes in vitro. When cultured for 10 days under differentiation-permissive conditions (details in Materials and Methods), hTERT + Bmi1 H#1 and H#2 polyclonal populations spontaneously differentiate into myotubes (Fig 3J). Similarly, hTERT + Bmi1 clones spontaneously differentiate into myotubes, albeit at lower frequency (Fig EV2F). Previous observations lead to the conclusion that human mesoangioblasts have a variable myogenic potency (Bonfanti et al, 2015), which decreases with passages in culture and which can be efficiently rescued by the expression of the myogenesis regulator MyoD (Morosetti et al, 2006; Tedesco et al, 2011). To test whether this was also the case for hTERT + Bmi1 clones and to minimise myogenic variability, reversibly immortalised mesoangioblasts were transduced with a lentiviral vector carrying the MyoD cDNA, fused with the estrogen receptor (ER) in order to activate its expression upon tamoxifen administration (MyoD-ER; Kimura et al, 2008; Tedesco et al, 2012). As a result, few days after tamoxifen activation of MyoD-ER, reversibly immortalised human mesoangioblasts were able to differentiate into skeletal myotubes with higher efficiency (Figs 3J

and EV2F). Taken together, these results demonstrate that expression of hTERT and Bmi1 in human mesoangioblasts enables bypassing replicative senescence via telomere maintenance and cell cycle progression without interfering with either spontaneous or induced myogenic differentiation.

### Immortalised human mesoangioblasts are not tumorigenic and their immortalisation can be reverted by Cre recombinase and ganciclovir

High proliferation rate and maintenance of telomerase expression are associated with cell transformation (Stewart et al, 2002). To rule out that hTERT + Bmi1-mediated immortalisation of human mesoangioblasts could induce cell transformation and tumorigenic conversion, cell–cell contact inhibition of proliferation and serum/growth factor dependence were tested in vitro. In both experimental sets, proliferation was evaluated as BrdU incorporation rate. Results showed that both clonal and polyclonal reversibly immortalised mesoangioblasts were sensitive to serum and growth factor withdrawal (Fig 4A), as well as to cell contact inhibition of proliferation (Fig 4B), indicating that lentiviral hTERT + Bmi1-mediated reversible immortalisation does not lead to cell transformation. Moreover, karyotype analysis performed on immortalised cells showed a normal diploid chromosomal content (Fig 4C). To confirm this data, in vivo tumorigenic assays were performed injecting clonal and polyclonal hTERT + Bmi1 mesoangioblasts subcutaneously into immunodeficient scid/beige mice (N = 4 per cell population). No tumours developed in any of the mice injected with immortalised mesoangioblasts after 12 months of follow-up (N = 20; Table EV1).

Overexpression of hTERT provides extension of cell proliferation while maintaining a stable chromosomal content (Cudre-Mauroux et al, 2003; Zhu et al, 2007; Mamchaoui et al, 2011; Robin et al, 2015). Nonetheless, lentiviral vectors carrying the immortalising cassette were designed to guarantee reversibility of

immortalisation (Salmon *et al*, 2000) to prevent potential adverse events related to the reconstitution of telomerase activity and guarantee a maximum level of safety of our platform (Bernardes de Jesus & Blasco, 2013; Chiodi & Mondello, 2016; Terali & Yilmazer, 2016). Firstly, loxP sites were positioned to flank hTERT and Bmi1 cassettes, enabling Cre-mediated excision of integrated transgenes. Secondly, hTERT and the HSV1-TK cDNAs were connected by an internal ribosome entry site (IRES) sequence to allow human cytomegalovirus (CMV) promoter-mediated concomitant transcription, so that cells expressing the

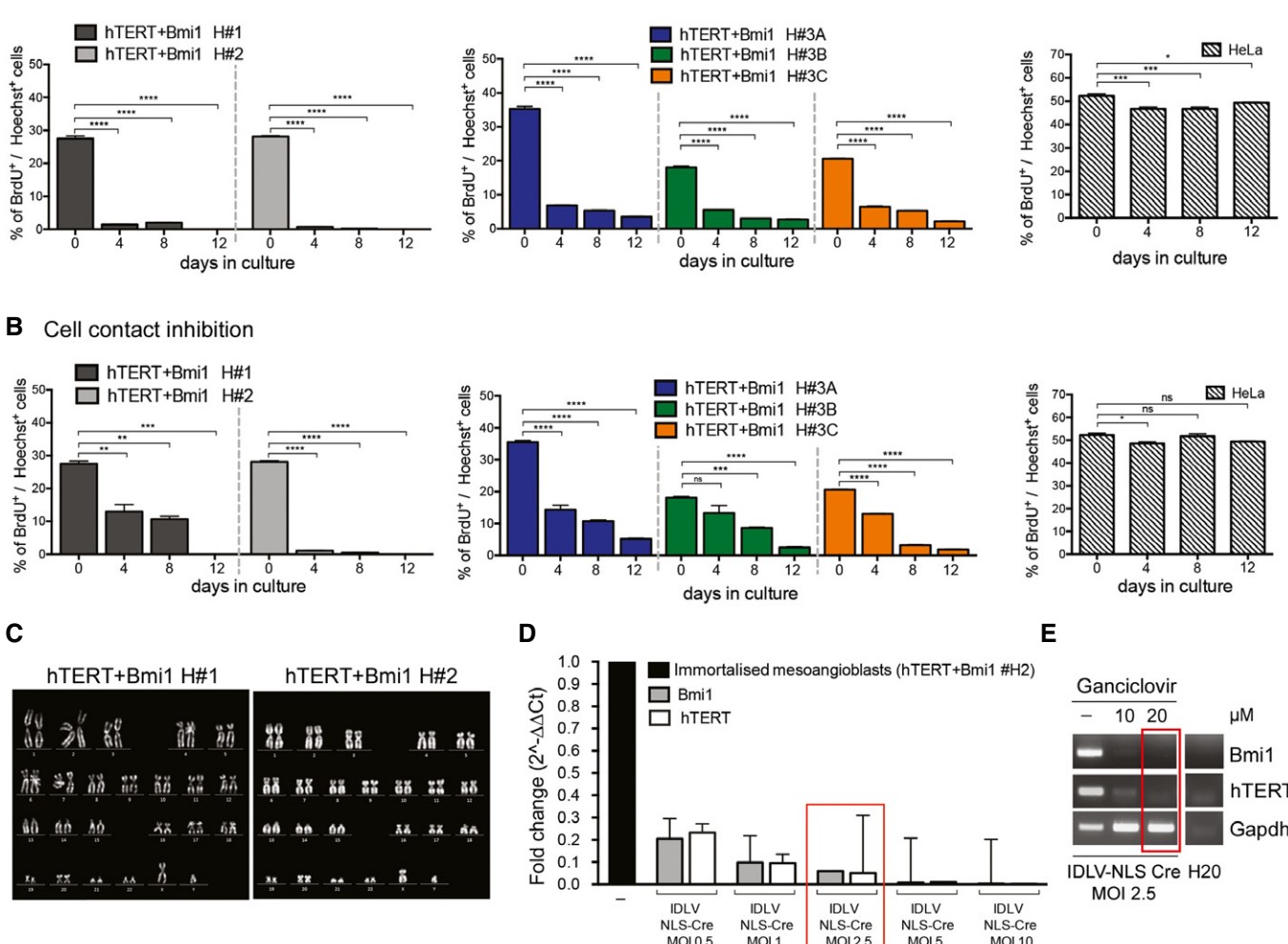

**Figure 4. Safety profile of reversibly immortalised human mesoangioblasts.**

A   *In vitro* serum and growth factor dependence of hTERT + Bmi1 polyclonal populations (H#1 and H#2, left graphs, *n* = 2 for each groups) and hTERT + Bmi1 clones (H#3A, H#3B and H#3C, right graphs, *n* = 4). HeLa cells: positive control (*n* = 4). BrdU incorporation rate was calculated as the % of BrdU-positive cells on total number of nuclei in 1 h time. Data plotted as means ± SEM. *$P$ = 0.0314, ***$P$ = 0.0002, ****$P$ < 0.0001, one-way analysis of variance (ANOVA) with Tukey's *post hoc* post-test.

B   Cell contact inhibition assays of hTERT + Bmi1 polyclonal populations (H#1 and H#2, left graphs, *n* = 2 for each groups) and hTERT + Bmi1 clones (H#3A, H#3B and H#3C, right graphs, *n* = 4). HeLa cells: positive control (*n* = 4). BrdU incorporation rate was calculated as the % of BrdU-positive cells on total number of nuclei. Data plotted as means ± SEM. *$P$ = 0.0190 (HeLa, 0 vs. 4), **$P$ = 0.0035 (H#1, 0 vs. 4), **$P$ = 0.0020 (H#1, 0 vs. 8), ***$P$ = 0.0003 (H#1, 0 vs. 12), ***$P$ = 0.0006 (H#3B, 0 vs. 8), **** $P$ < 0.0001, ns = 0.0627 (H#3B, 0 vs. 4), ns = 0.9643 (HeLa, 0 vs. 8), ns = 0.0790 (HeLa, 0 vs. 12), one-way analysis of variance (ANOVA) with Tukey's *post hoc* post-test.

C   Karyotype analysis of immortalised mesoangioblast polyclonal populations.

D   qRT for hTERT and Bmi1 transgenes in immortalised mesoangioblasts (H#2) 2 weeks after transduction with different multiplicity of infection (MOI 0.5, 1, 2.5, 5 and 10) of IDLV NLS-Cre. Gapdh was used as housekeeping gene. Immortalised mesoangioblasts have been used as reference (=1, black bar). Data plotted as means ± SEM (*n* = 2). Red box: IDLV NLS-Cre MOI 2.5 immortalised mesoangioblasts used for further experiments with ganciclovir.

E   hTERT and Bmi1 PCRs on MOI 2.5 IDLV NLS-Cre-transduced immortalised mesoangioblasts treated with 10 and 20 μM ganciclovir. Positive control: untreated (−) immortalised mesoangioblasts.

Source data are available online for this figure.

HSV1-TK would undergo apoptosis in the presence of the antiviral drug ganciclovir (Fillat *et al*, 2003). Hence, cells that escape Cre-mediated excision of the floxed hTERT or Bmi1 cDNAs will keep transcribing the TK cDNA, thus remaining sensitive to eradication by ganciclovir. An integrase-defective lentiviral vector expressing a nuclear localisation signal Cre recombinase (IDLV NLS-Cre) was then used to excise the immortalising transgenes.

To test the feasibility of the excision system, reversibly immortalised mesoangioblasts (hTERT + Bmi1 H#2) were transduced with different concentrations of IDLV NLS-Cre for 24 h [multiplicity of infection (MOI) 0.5, 1, 2.5, 5 and 10]. Cells were kept in culture for 2 weeks to allow dilution of excised transgenes, which might be otherwise detected by DNA analyses. Quantitative real-time PCR analysis of hTERT and Bmi1 transgenes was performed on Cre-transduced, immortalised mesoangioblasts. Dose-dependent reduction in both hTERT and Bmi1 was observed in all samples transduced with IDLV NLS-Cre, ranging from 75% (MOI 0.5) up to 99% (MOI 10; Fig 4D). These data demonstrate that IDLV NLS-Cre recombinase efficiently excises loxP-flanked hTERT and Bmi1 transgenes in human mesoangioblasts. Similar levels of hTERT and Bmi1 transgenes excision were confirmed by transduction of immortalised myoblasts with IDLV NLS-Cre (Appendix Fig S1B). To investigate whether the small percentage of cells that fail to excise the immortalising cassettes might have a growth advantage in the absence of ganciclovir counter-selection, we studied Bmi1 and hTERT expression levels and cell proliferation at different time points after IDLV NLS-Cre transduction (4, 5, 6, 7 and 8 weeks). At 4 weeks, Bmi1 and hTERT levels were raised from 5 to 6% (MOI 2.5 IDLV NLS-Cre; Fig 4D, red box) up to 40–50% (MOI 2.5 IDLV NLS-Cre; Appendix Fig S1C) and continued to increase up to 8 weeks, reaching the same levels of hTERT and Bmi1 as immortalised mesoangioblasts not treated with Cre recombinase (Appendix Fig S1C). Proliferation rate of IDLV NLS-Cre mesoangioblasts at 4 weeks from IDLV NLS-Cre transduction was lower than immortalised mesoangioblasts (42.3 ± 2.9% vs. 33.5 ± 1.3%), consistent with a reduction in hTERT- and Bmi1-positive cells (Appendix Fig S1D). The decrease in proliferation was then restored at 8 weeks after IDLV NLS-Cre (33.4 ± 2.1% vs. 34.4 ± 1.3%; Appendix Fig S1D). These results showed that

in the absence of counter-selection, cells that have retained the immortalising genes have a moderate but significant growth advantage. As a result, we administered ganciclovir counter-selection within 2 weeks of transduction with IDLV NLS-Cre to eliminate cells that escaped Cre-loxP-mediated transgene excision. As only cells carrying both immortalising transgene are able to undergo effective immortalisation (Cudre-Mauroux *et al*, 2003), the HSV1-TK cDNA was transcriptionally linked only with hTERT using an IRES sequence; this reduced the transgene size and the likelihood of a bystander effect elicited by a few TK-expressing cells inducing cell death of TK-negative neighbouring cells (Freeman *et al*, 1993; Denning & Pitts, 1997). For these experiments, we selected MOI 2.5 IDLV NLS-Cre-transduced immortalised mesoangioblasts (Fig 4D, red box), as 5–6% of them escaped Cre-mediated hTERT and Bmi1 excision. IDLV NLS-Cre-transduced immortalised mesoangioblasts were treated with 10 or 20 μM ganciclovir for 3 weeks and then PCRs for hTERT and Bmi1 transgenes were performed. As shown in Fig 4E, IDLV NLS-Cre-transduced immortalised mesoangioblasts were negative for both transgenes after treatment with 20 μM ganciclovir. Taken together, these results demonstrate that immortalisation of human mesoangioblasts with hTERT and Bmi1 (i) does not induce cell transformation, (ii) does not lead to tumorigenic conversion, (iii) can be safely reverted by transgene excision via expression of Cre recombinase and ganciclovir treatment.

## Reversible immortalisation enables DYS-HAC-mediated genetic correction of DMD mesoangioblasts

The above-described data demonstrate that reversible immortalisation of healthy donor human mesoangioblasts enables safe bypassing of replicative senescence. The same platform was then utilised to reversibly immortalise mesoangioblasts isolated from skeletal muscle biopsies of DMD patients. DMD mesoangioblasts were successfully transduced with hTERT and Bmi1 lentiviral vectors (Fig 5A and B), showed extension of their proliferative ability (Fig 5C), and were not tumorigenic (*N* = 3; Table EV1). Thus, we used reversibly immortalised DMD (riDMD) mesoangioblasts as recipient myogenic progenitor cells for the newly generated

**Figure 5. Immortalisation of DMD mesoangioblasts and DYS-HAC2 transfer.**

A   Phase contrast morphology of DMD human mesoangioblasts before (CTR) and after transduction with lentiviral vectors carrying the immortalising cDNAs (hTERT + Bmi1). Scale bar: 100 μm.

B   Bmi1 and hTERT PCRs on DMD and hTERT + Bmi1 reversibly immortalised DMD (riDMD) mesoangioblasts.

C   DMD and riDMD mesoangioblasts population doubling curves. Data are expressed as means ± SEM (*n* = 2). **P = 0.008, unpaired two-tailed *t*-test performed on last time point.

D   PCR analysis on riDMD(DYS-HAC2) mesoangioblasts clones obtained after MMCT. Primers were designed to analyse HAC-specific sequences (Neo and 3′) and dystrophin exons (exons 46 and 50) deleted in the DMD cells used for the study (deletion from exon 45 to 50). CHO(DYS-HAC2)-7: positive control; healthy donor human mesoangioblasts: negative control for HAC sequences and positive control for dystrophin exons; riDMD mesoangioblasts: negative control for exons spanning from 45 to 50. Exon 23 was used as internal control and normaliser.

E   Upper panel: FISH analyses of riDMD(DYS-HAC2) mesoangioblast clones. White arrowheads and insets: single episomal copy of DYS-HAC2. Red: rhodamine-p11-4 human alpha satellite (centromeres of chromosome 13 and 21, hChr 13/21(cen)); green: dystrophin FITC-DMD-BAC RP11-954B16. Scale bar: 3 μm. Lower panel: karyotype analysis of riDMD(DYS-HAC2) mesoangioblast clones.

F   Left: bar graph showing results of *in vitro* transmigration assay of riDMD(DYS-HAC2) mesoangioblast clones (*N* = 4, *n* = 4), riDMD mesoangioblast parental population (*n* = 4), human mesoangioblasts (*n* = 3) and human myoblasts (*n* = 3). Data plotted as means ± SEM. *P = 0.0378, **P = 0.0081 [riDMD(DYS-HAC2) clones], **P = 0.0032 (riDMD mesoangioblasts), one-way analysis of variance (ANOVA) with Tukey's *post hoc* post-test. Right: fluorescence images of a riDMD (DYS-HAC2) #B clone. Human mesoangioblasts and myoblasts were labelled with 6-carboxyfluorescein diacetate (green). Scale bar: 250 μm.

Source data are available online for this figure.

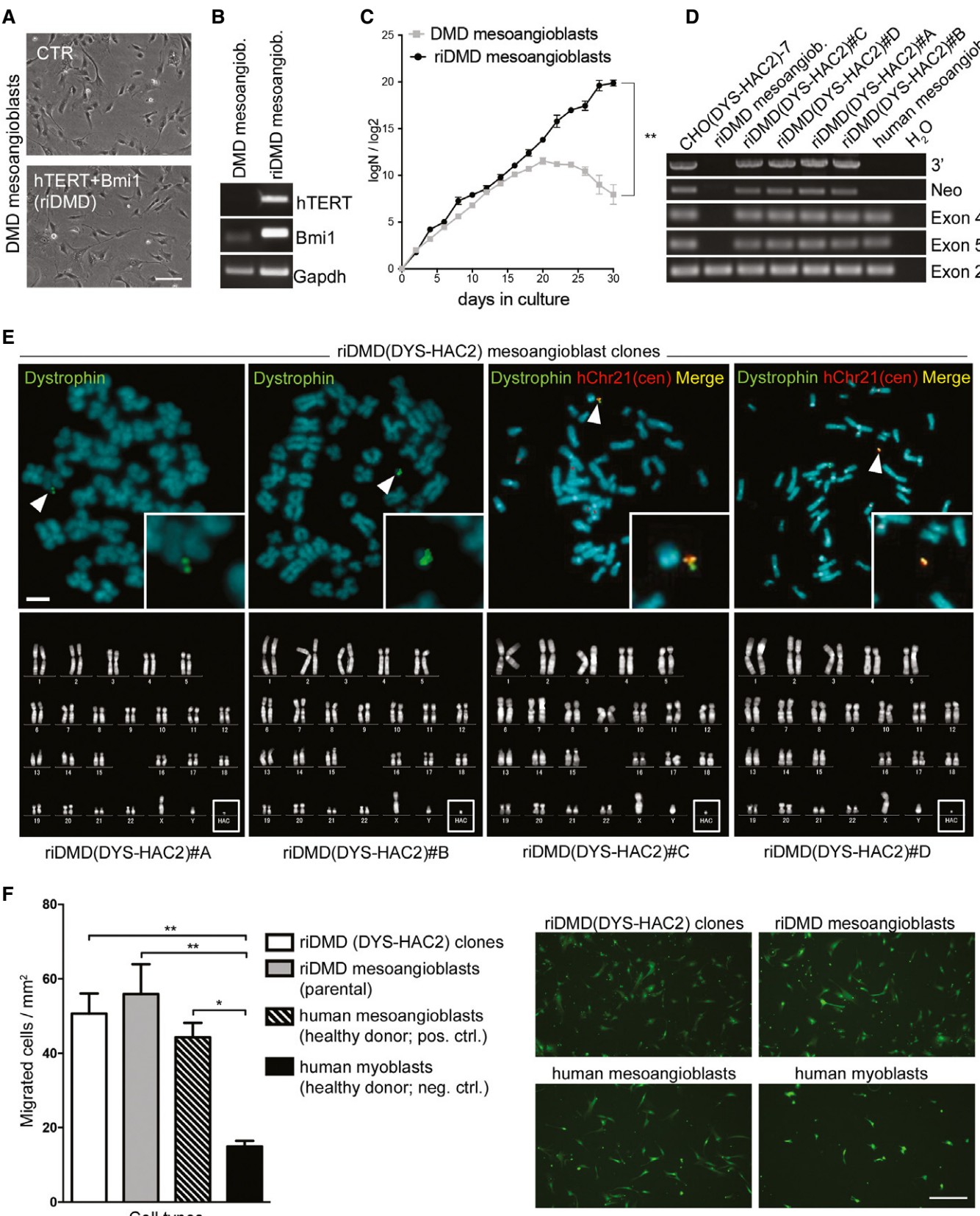

Figure 5.

DYS-HAC2 (Fig 1), as previously done with riDMD myoblasts (Fig 2). Two different MMCT techniques, as well as two different HAC donor cell types, were used [CHO(DYS-HAC2)-7 and A9(DYS-HAC2)-9 clones, additional information in Appendix Supplementary Methods]. Following DYS-HAC2 transfer and subsequent G418 selection, four riDMD(DYS-HAC2) mesoangioblast clones were obtained, namely riDMD(DYS-HAC2)#A, #B, #C and #D. PCR analyses of specific HAC sequences and dystrophin exons deleted in the parental cells (from exons 45 to 50; Fig 5D) together with FISH analyses (Fig 5E, upper panel), confirmed single and episomal DYS-HAC2 presence. Additional PCRs for human (human actinin)- and hamster-specific genes (furin and NV1) ruled out CHO contamination in the clones (Fig EV3A). Karyotype analysis showed normal chromosomal content plus DYS-HAC2 (46 XY + DYS-HAC2; Fig 5E, bottom panel). Tumorigenic assays in *scid/beige* immunodeficient mice ($N = 6–8$ per cell population) showed the absence of tumour formation and confirmed the safety profile of riDMD(DYS-HAC2) mesoangioblasts *in vivo* ($N = 27$; Table EV1). Phase contrast microscopy showed morphology comparable with parental cells (Fig EV3B) and population doubling curves demonstrated maintenance of proliferative potential (Fig EV3C and D). One of the key features of mesoangioblasts is their ability to interact with endothelial cells and cross the blood vessel wall when delivered systemically (Giannotta *et al*, 2014). To assess preservation of this property, riDMD(DYS-HAC2) mesoangioblasts were subjected to an *in vitro* transmigration assay (Giannotta *et al*, 2014; Bonfanti *et al*, 2015). Human umbilical vein endothelial cells (HUVECs) were grown as confluent monolayers on gelatin-coated filters and fluorescently labelled mesoangioblasts were allowed to transmigrate for 10 h (Fig 5F). Quantification of the assay showed that riDMD(DYS-HAC2) mesoangioblasts and the immortalised parental population (riDMD mesoangioblasts) transmigrated to the same extent than normal human mesoangioblasts ($50.7 \pm 6.2$, $55.9 \pm 9.3$, $44.3 \pm 6.9$ cells/mm$^2$, respectively) and significantly more than human myoblasts ($15 \pm 1.4$ cells/mm$^2$; Fig 5F). Therefore, immortalisation and DYS-HAC transfer do not interfere with the migration potential of human mesoangioblast.

**Reversibly immortalised, DYS-HAC-corrected, DMD progenitors engraft murine skeletal muscle upon transplantation**

To avoid immune reaction against human cells, immunodeficient *NOD/scid/gamma* (NSG) mice were first used as recipient hosts for xenotransplantation to test survival and engraftment of reversibly immortalised human myoblasts. NSG tibialis anterior muscles were subjected to cryoinjury to induce regeneration and then injected intramuscularly with $10^6$ riDMD(DYS-HAC2) myoblasts (clone #α). Three weeks later, muscles were collected and immunofluorescence analysis for lamin A/C (which marks the nuclear lamina of human donor cells) and laminin (marking extracellular matrix) on muscle sections showed that DYS-HAC-corrected myoblasts engrafted regenerating murine skeletal muscle and transcribed human dystrophin from the HAC (Fig 6A). We then tested riDMD (DYS-HAC2) mesoangioblasts in the same set-up (cryoinjured NSG mice), injecting $10^6$ MyoD-ER-transduced cells from clone #C. Also in this case, muscle sections showed that DYS-HAC-corrected DMD mesoangioblasts engrafted regenerating murine skeletal muscle (Fig 6B and Appendix Fig S2). To further investigate the potential of

reversibly immortalised mesoangioblasts to engraft in a DMD mouse model, dystrophic immunodeficient *scid/mdx* mice were used for proof-of-principle xenotransplantation. Reversibly immortalised MyoD-ER-transduced healthy donor mesoangioblasts (hTERT + Bmi1 H#2) or DYS-HAC-corrected DMD mesoangioblasts [riDMD (DYS-HAC2)#A] were injected into tibialis anterior muscles of *scid/mdx* mice. Muscles were explanted after 3 weeks, and immunofluorescence analyses showed the presence of dystrophin-positive myofibres containing human lamin A/C-positive nuclei (Fig 6C and D). Moreover, the aforementioned findings were validated performing a heterotopic transplantation assay to assess cell-autonomous dystrophin production from donor human cells. To this aim, $10^6$ riDMD(DYS-HAC2) myoblasts (clone #α) were injected in subcutaneous Matrigel plugs in NSG-immunodeficient mice [as recently reported (Sacchetti *et al*, 2016)]; plugs were explanted and analysed 14 days after injection. Notably, MyHC-positive myotube-like structures double positive for lamin A/C and dystrophin were identified in all mice, confirming dystrophin production from HAC-corrected donor human cells (Fig 6E and Appendix Fig S3; please note that dystrophin could only be produced by HAC-corrected cells, as no host myofibres provided background signal). Taken together, these results demonstrate engraftment and differentiation of DYS-HAC-corrected DMD muscle progenitors *in vivo*.

**Development of a next-generation, multifunctional, reversibly immortalising DYS-HAC**

Our data established feasibility and safety of the lentiviral reversibly immortalising platform for HAC transfer in primary human muscle progenitors. To simplify this procedure, we combined all required gene functions into a single HAC capable of simultaneously delivering genomic integration-free reversible immortalisation, genetic correction, inducible differentiation and controllable cell death (Fig 7A and Appendix Fig S4). Construction of this next-generation HAC (referred to as DYS-HAC4) involved engineering of several sequences by gene synthesis (Fig 7A and Appendix Fig S4 for a detailed map), including (i) the 2.4-Mb dystrophin locus for complete genetic correction, (ii) a hTERT and Bmi1 immortalising cassette with an elimination system via CreERT2/loxP and negative selection by TK, (iii) a clinically tested safeguard system based upon inducible Caspase 9 (iCasp9; Di Stasi *et al*, 2011), prompting apoptosis upon administration of the drug AP1903, (iv) an inducible myogenic differentiation system based upon human MYOD nuclear translocation after tamoxifen administration (MYOD-ERT2; Tedesco *et al*, 2011; Maffioletti *et al*, 2015) to rescue myogenic capacity in cells that might lose it after prolonged or high-density culture, (v) a codon-optimised human dystrophin cDNA (huDYSco) under the control of the synthetic Spc5-12 muscle-specific promoter (details in Appendix Fig S5; Li *et al*, 1999; Loperfido *et al*, 2015) to increase dystrophin expression, thus maximising the therapeutic effect upon transplantation and fusion with multinucleated, dystrophin-negative myofibres.

DYS-HAC4 was engineered with a two-step protocol by site-specific integration of plasmids p17 and pP-ΔHR (Fig 7A, yellow and blue lines, respectively) into DYS-HAC2 (Fig 1A and black line in Fig 7A; details in Appendix Fig S4). Firstly, the p17 plasmid carrying huDYSco and iCas9-MYOD-ERT2 cassettes was transferred into CHO cells and integrated into DYS-HAC2 via Cre/loxP

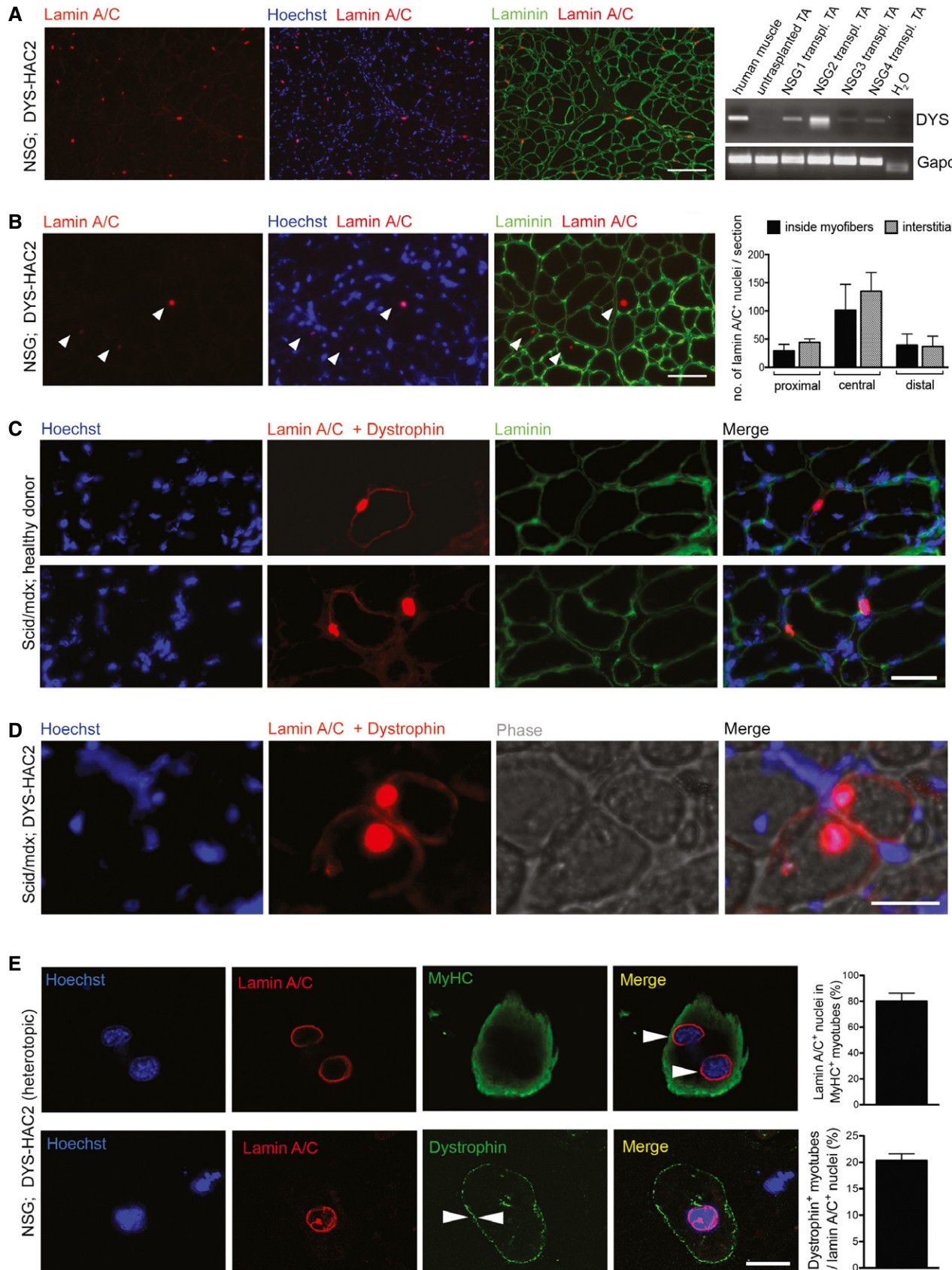

Figure 6.

◀

**Figure 6. Transplantation of reversibly immortalised human muscle progenitors in immunodeficient mice.**

A    Immunofluorescence images (representative) showing donor-derived lamin A/C-positive nuclei (red) in a cryoinjured tibialis anterior muscle of an NSG mouse after intramuscular injection of $10^6$ riDMD(DYS-HAC2) myoblasts (clone #α). Green: laminin; blue: Hoechst. Scale bar: 100 μm. Right panel: RT–PCR analysis of the same transplanted muscles showing human dystrophin mRNA expression in the tibialis anterior (TA) muscles of four NSG mice (human muscle: positive control; untransplanted TA: negative control).

B    Immunofluorescence images showing donor-derived lamin A/C-positive nuclei (red) in a cryoinjured tibialis anterior muscle of an NSG mouse after intramuscular injection of $10^6$ MyoD-ER-transduced riDMD(DYS-HAC2) mesoangioblasts [riDMD(DYS-HAC2)#C]. Green: laminin; blue: Hoechst. White arrowheads: human lamin A/C-positive nuclei. Scale bars: 75 μm. Right bar graph: quantification of lamin A/C-positive nuclei in the same experiment. Data plotted as means ± SEM (N = 3; additional information in Appendix Fig S2).

C    Immunofluorescence of frozen section of a scid/mdx mouse tibialis anterior muscle upon intramuscular injection with $10^6$ MyoD-ER-transduced healthy donor immortalised mesoangioblasts (hTERT + Bmi1 H#2). Red: lamin A/C and dystrophin; green: laminin; blue: Hoechst. Scale bars: 50 μm.

D    Immunofluorescence on a frozen section of a scid/mdx mouse tibialis anterior muscle upon intramuscular injection with $10^6$ MyoD-ER-transduced riDMD(DYS-HAC2) mesoangioblasts [riDMD(DYS-HAC2)#A]. IM: intramuscular. Red: lamin A/C and dystrophin; blue: Hoechst. Scale bars: 50 μm.

E    Heterotopic subcutaneous transplantation assay. Upper panel: confocal microscopy pictures showing a representative myosin heavy chain (MyHC)-positive (green), myotube-like structure containing human lamin A/C-positive nuclei (red) upon immunofluorescence staining of Matrigel plugs ($10^6$ cells were injected subcutaneously 2 weeks earlier in four immunodeficient NSG mice; arrowheads highlight multinucleation). Lower panel: confocal microscopy pictures showing another representative myotube-like structure from the same experiment immunostained positively for both lamin A/C (red) and dystrophin (green). Arrowheads highlight dystrophin staining pattern. Bar graphs quantify respective panels. Four mice were transplanted, all showed engraftment, and Matrigel plugs from three randomly selected animals were quantified. A minimum of 130 human nuclei per plug were counted (in nine randomly selected high-power fields). Data plotted as means ± SEM (N = 3). Additional details in Appendix Fig S3. Scale bar: 15 μm.

Source data are available online for this figure.

recombination, generating an intermediate DYS-HAC3 (Step 1, Appendix Fig S4). The Cre-mediated recombination between the 5′HPRT-lox71 site on DYS-HAC2 and the loxJTZ17-3′HPRT site on p17 resulted in HPRT gene reconstruction and HAT resistance (Kazuki et al, 2011). DYS-HAC3-containing CHO cells were then selected in HAT-supplemented medium. Subsequently, the pP-ΔHR plasmid carrying the immortalising cassette was transferred into CHO(DYS-HAC3) cells and integration into DYS-HAC3 was achieved via Bxb1 attB/attP sites and the Bxb1 integrase system (Yamaguchi et al, 2011; Step 2, Appendix Fig S4), resulting in L-histidinol dihydrochloride resistance gene (hisD) reconstitution (Tucker & Burke, 1996). Selection of CHO cells with recombinant DYS-HAC3 and pP-ΔHR plasmid was obtained by culturing cells with L-histidinol dihydrochloride. Successful construction of DYS-HAC4 was then confirmed by PCR analysis of genomic dystrophin sequences in selected CHO(DYS-HAC4) clones (Fig 7B–D). CHO(DYS-HAC4) clones #19 and #20 were investigated further for the presence of the novel cassettes and CHO(DYS-HAC4) clone #19 was shown to be positive for all relevant sequences, that is huDYSco, iCaspase9, hTERT, Bmi1 and the recombination junction with Bxb1 integrase (Fig 7C and D). FISH analysis of CHO(DYS-HAC4)#19 revealed that 90% of interphase cells (n = 100) contained DYS-HAC4 and 20% metaphases (n = 20) maintained a single episomal copy of DYS-HAC4 (Fig 7E). Finally, RT–PCR revealed expression of all new transgenes (Fig 7F). Detection of huDYSco expression in undifferentiated cells could be due to interspecific issues (i.e. human sequence in hamster background) and/or occasional leakiness of the Spc5-12 promoter in tissues other than skeletal muscle (Rincon et al, 2015). The total size of this novel synthetic HAC is in the range of 7 Mb.

## Discussion

Here, we describe a novel stepwise strategy to enable genetic correction of clinically relevant, human skeletal muscle-derived progenitors for gene and cell therapy. Reversibly immortalising lentiviral vectors proved to be safe and efficacious in extending the proliferative capacity of DMD muscle progenitors, enabling

them to withstand transfer of a novel HAC containing the whole dystrophin locus and the subsequent large-scale clonal cell expansion required for bioprocessing of an advanced therapy medicinal product. Notably, starting from a limited cell number ($3 \times 10^4$), we obtained within 31 days a number of cells potentially sufficient to treat a DMD paediatric patient (in the range of $10^9$ cells; Cossu et al, 2015). Importantly, this work has been conducted in two independent laboratories (in the UK and in Japan) and validated using (i) two distinct cell types, (ii) healthy and dystrophic genotypes, (iii) five different donors, (iv) polyclonal and clonal analyses.

Previous work by Trono and colleagues showed that combined expression of hTERT and Bmi1 bypassed replicative senescence in myoblasts (Cudre-Mauroux et al, 2003). Taking advantage of the availability of DMD immortalised myoblasts, we successfully transferred DYS-HAC2 in these cells. However, although myoblast transplantation is promising for localised forms of muscular dystrophy (Perie et al, 2014), their use in widespread muscle disorders such as DMD is limited. Hence, we selected mesoangioblasts to take advantage of their migration ability and transferred DYS-HAC2 by extending their proliferation potential using the same immortalisation platform. hTERT-TK and Bmi1 lentiviral vectors used in this study were originally designed to enable reversion of the immortalised status in the presence of Cre recombinase and ganciclovir (Salmon et al, 2000). Translation of the hTERT and Bmi1 immortalising platform to DMD mesoangioblasts proved that this approach was efficient at extending cell proliferation and enabled clonal DYS-HAC transfer and subsequent amplification. The safe use of lentiviral vectors to mediate integration and excision of the immortalising cassettes utilised in this study is supported by recent gene therapy clinical studies (Naldini, 2015). Additionally, our observations are in keeping with a number of reports showing that cellular senescence can be overcome by telomerase reconstitution in combination with Bmi1 or CDK4 expression, without compromising genomic stability, migration ability, myogenic potency or capacity to engraft skeletal muscle (Cudre-Mauroux et al, 2003; Zhu et al, 2007; Shiomi et al, 2011; Robin et al, 2015). Even though we did not observe Pax7 expression in immortalised cells, it is unlikely that

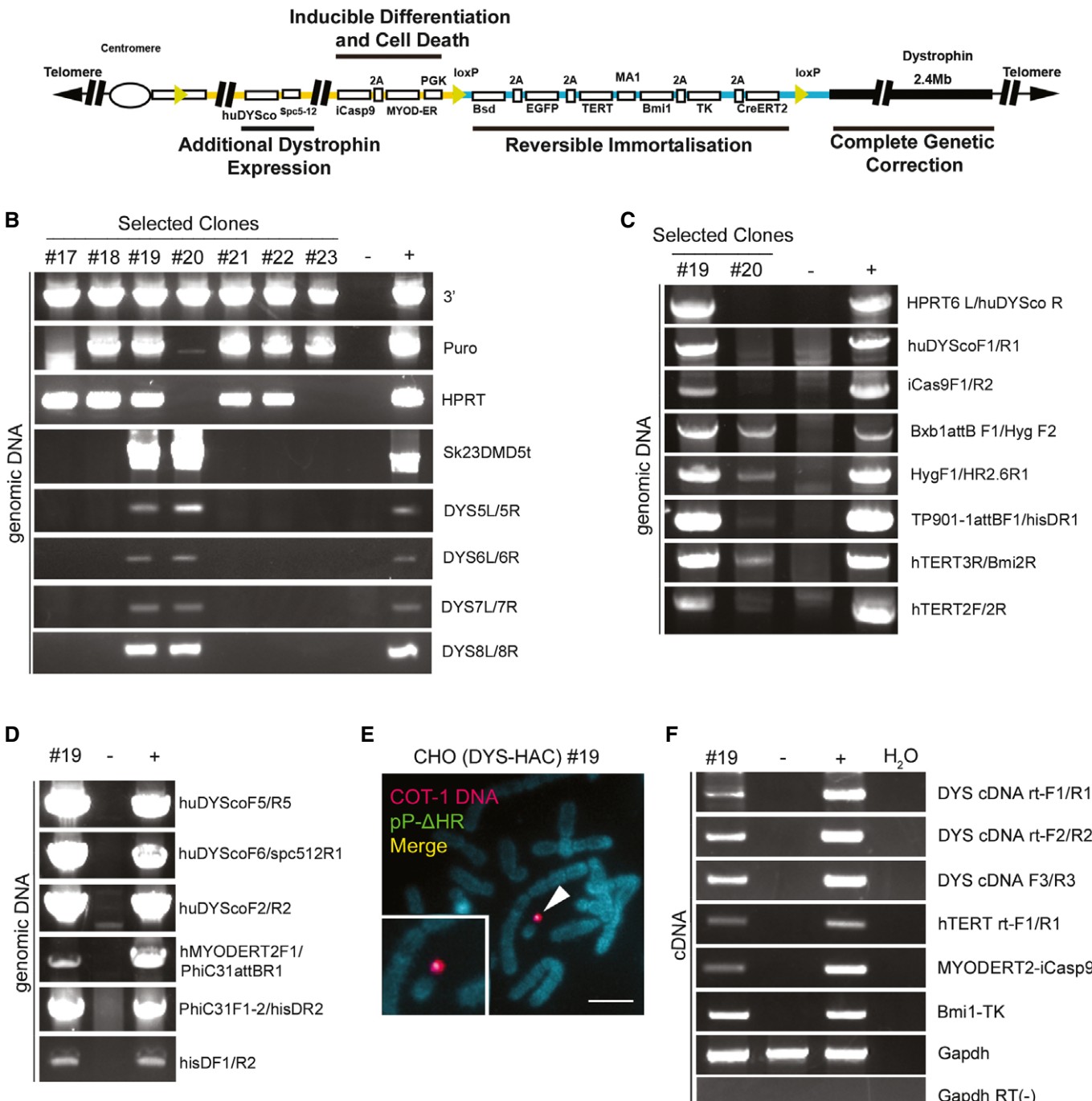

**Figure 7.**

immortalising vectors caused this, as Bmi1 plays a key role in supporting skeletal muscle regeneration and maintaining the satellite cell pool (Robson *et al*, 2011; Di Foggia *et al*, 2014; Dibenedetto *et al*, 2017). This might be due to extensive passaging *in vitro* or to the amplification of cells that might have already lost Pax7 expression during the initial expansion pre-transduction. Recent advances in human satellite cell purification (Xu *et al*, 2015) together with

immortalisation at early passages could improve this outcome. The same process could be less relevant for human mesoangioblasts, whose dependence on Pax7 is unclear. Variability of myogenic capacity could also be a consequence of high-density cultures during the MMCT technique necessary for HAC transfer. In this event, a contingency plan with MyoD-ER-induced rescue of differentiation has been described here.

**Figure 7.  Generation of a novel, synthetic, multifunctional DYS-HAC.**

A    Schematic diagram of DYS-HAC4, generated by integration of plasmids p17 (yellow line) and pP-ΔHR (blue line) into DYS-HAC2 (Fig 1A) as shown also in Appendix Fig S4. Four functional cassettes are present: (i) dystrophin locus (2.4 Mb) for complete genetic correction, (ii) hTERT and Bmi1 immortalising cassette under control of MA1 bidirectional promoter (Amendola et al, 2005) and floxed by loxP sites to be eliminated via Cre-loxP recombination system. Excision of the immortalising cassette is monitored by EGFP expression, response to blasticidin (Bsd) resistance and sensitivity to ganciclovir (TK), (iii) codon-optimised human dystrophin (huDYSco, 11.1 kb) under control of the Spc5-12 promoter to increase dystrophin expression, (iv) inducible Caspase 9 (iCas9) and human MYOD-ERT2 under a PGK promoter for controllable cell death and myogenic differentiation, respectively.

B    PCR analyses of selected CHO(DYS-HAC4) clones confirming the presence of DNA sequences derived from DYS-HAC2 (DYS-HAC backbone detected with 3′, Puro and Sk23/DMD5t primers, Cre-lox71/loxJTZ17 recombination detected with HPRT primers, genomic dystrophin sequence detected using DYS5L/5R, DYS6L/6R DYS7L/7R and DYS8L/8R primers).

C, D   PCR analyses of CHO(DYS-HAC4) clones #19 and #20 showing the presence of all relevant novel sequences confirming insertion of plasmid p17 and pP-ΔHR.

E    FISH analysis of CHO(DYS-HAC4) clone #19 showing episomal presence of DYS-HAC4 in single copy (red: rhodamine-human COT-1 DNA; green: FITC-Plasmid pP-ΔHR containing the immortalising cassette. Scale bar: 5 μm.

F    RT–PCR analysis showing expression of huDYSco, immortalising cassette (hTERT and Bmi1), MYOD-ERT2 and iCaspase 9 in CHO(DYS-HAC4) clone #19. For all PCRs, CHO-K1 cells were used as negative control and CHO(DYS-HAC4) parental population as positive control.

Source data are available online for this figure.

In this work, we have also advanced chromosome engineering for *ex vivo* gene therapies. We firstly generated a novel DYS-HAC, namely DYS-HAC2, devoid of potentially immunogenic genes and therefore one step closer to a possible clinical-grade product compared to the previous DYS-HAC (Hoshiya et al, 2009). Additionally, we have developed a next-generation HAC capable of delivering genomic integration-free reversible immortalisation, genetic correction, additional dystrophin expression, inducible differentiation and controllable cell death—all simultaneously in an episomal vector. To our knowledge, this novel HAC represents the largest and possibly most complex gene therapy vector developed to date: future studies will assess its function *in vitro* and *in vivo* after transfer into different human myogenic cells, including timing and dynamics of clonal HAC-mediated immortalisation in primary cells.

Alternative mutation-specific strategies are currently being developed for DMD (Foster et al, 2012; Bengtsson et al, 2016). Among these, antisense oligonucleotide-mediated exon skipping appears promising and ataluren has recently received market authorisation for nonsense mutations; nonetheless, recent clinical trials highlight the need for further studies to clarify clinical efficacy (Lu et al, 2014; Voit et al, 2014; Godfrey et al, 2017; McDonald et al, 2017). However, these strategies cannot be applied to a significant number of DMD mutations in functional domains or to large deletions (Aartsma-Rus et al, 2009). "Permanent" exon skipping via adeno-associated viral vector (AAV)-mediated gene editing also appears to be a promising option (Long et al, 2016; Nelson et al, 2016; Tabebordbar et al, 2016; Young et al, 2016); however, immune reactions against AAVs (Boisgerault & Mingozzi, 2015) and the editing enzyme (which usually is of prokaryotic origin) plus off-target effects might delay clinical translation. Although recent work provides encouraging *in vitro* evidence of lentiviral full-length dystrophin transfer into human myoblasts (Counsell et al, 2017), HAC-mediated gene therapy offers the chance to cover the entire spectrum of DMD mutations while preserving all the complex dystrophin regulatory elements (Muntoni et al, 2003). Additionally, safety profile and standard operating procedures for manufacturing both cell populations used in this study have been already developed in allogeneic and autologous clinical-grade conditions (Perie et al, 2014; Cossu et al, 2015), thus paving the way for future clinical translation of genetically corrected cells.

Although iPS cell-derived myogenic progenitors promise to overcome some of the hurdles of cell therapy for muscle disorders (reviewed in Loperfido et al, 2015), they are in very early and critical stage of pre-clinical translation: safety concerns will need to be fully addressed before usage in clinical settings requiring systemic or multiple administrations such as in DMD. Importantly, clinical translation of iPS cell-derived progenitors (e.g. Chal et al, 2015; Choi et al, 2016; Darabi et al, 2012; Loperfido et al, 2015; Tedesco et al, 2012) is likely to rely on prior successful clinical studies with their tissue-derived counterparts and technologies such as the one described here are likely to benefit this critical step.

Overall, we expect the translational journey of HAC-based cell therapy to be complex, but no more challenging than those of other gene therapy vectors (Tedesco, 2015). However, some hurdles still need to be overcome, such as the low efficiency of HAC transfer to recipient cells: new technologies to improve this aspect have been recently developed and are currently under investigation in our laboratories (Hiratsuka et al, 2015; Suzuki et al, 2016). Lastly, as much as DMD gene therapy relies on size/complexity of the dystrophin copy delivered to dystrophic cells, efficacy of cell therapy for DMD is linked to the amount of stem/progenitor cells able to reach and engraft the dystrophic muscles and their myogenic potency. Although we provided proof-of-principle evidence of engraftment and differentiation of HAC-corrected human progenitors *in vivo* (Fig 6), transplantation protocols will require optimisation to reach clinical efficacy. Recent work by our laboratories and others showed that it is possible to increase cell engraftment with consequent functional amelioration of the dystrophic phenotype (Hindi et al, 2013; Giannotta et al, 2014). Combination of these approaches (e.g. improvement of HAC transfer efficiency and augmented cell engraftment) could bring the DYS-HAC platform closer to clinical translation, although specific studies in small and large animal models will be required to define the extent of its safety and efficacy *in vivo*.

Taken together, this work provides the first evidence of HAC transfer into clinically relevant, dystrophic human muscle progenitors and describes the development of next-generation multifunctional artificial chromosomes for *ex vivo* gene therapy.

# Materials and Methods

### Cell isolation and culture

Primary cells were isolated from human healthy donor and DMD skeletal muscle biopsies as previously described (Tonlorenzi et al, 2007). Cells were subsequently harvested, suspended in 1% FBS, 2 mM EDTA PBS and incubated with CD56-PE (NCAM; Miltenyi Biotec) and ALP-FITC (Santa Cruz) antibodies for 30 min at 4°C. Human mesoangioblasts were isolated as alkaline phosphatase (ALP)-positive/CD56-negative fraction (Dellavalle et al, 2007) via FACS (DIVA Vantage, BD) and cultured in MegaCell Dulbecco's modified Eagle's medium (MegaCell DMEM, Sigma) containing 5% foetal bovine serum (FBS; Life Technologies), 2 mM glutamine, penicillin and streptomycin (100 IU + 0.1 mg/ml; Sigma), 1% non-essential amino acids (Sigma), 0.05 mM 2-mercaptoethanol (Gibco), 5 ng/ml of HGF (Sigma) and human basic FGF (Gibco). For in vitro skeletal muscle differentiation, human mesoangioblasts were plated on Matrigel-coated dishes, and once confluent, they were switched from MegaCell medium to DMEM (Sigma) 2% horse serum (Euroclone) plus 1% glutamine (Sigma), 1% penicillin/streptomycin (Sigma) and cultured for an additional 7–10 days before analysis. For MyoD-ER-induced differentiation of human mesoangioblasts and myoblasts previously transduced with tamoxifen-inducible MyoD-ER lentiviral vector, 4 OH-tamoxifen was added to culture medium following the previously published protocols (Gerli et al, 2014). Specific tamoxifen concentration and duration of pulse requires cell line-specific titration for optimal results.

Chicken DT40(DYS-HAC2) cells were maintained at 40°C in RPMI (Sigma) medium supplemented with 10% foetal bovine serum (FBS), 1% chicken serum, 50 μm 2-mercaptoethanol, 1% glutamine and 1% penicillin/streptomycin plus 1.5 mg/ml G418 (neomycin, Sigma). CHO(DYS-HAC2) cells were cultured in Ham's F-12 nutrient mixture (Sigma) with 10% FBS, 1% glutamine, 1% penicillin/streptomycin and 0.8 mg/ml G418. A9(DYS-HAC2) cells were maintained in DMEM plus 10% FBS, 1% glutamine, 1% penicillin/streptomycin and 0.8 mg/ml G418. riDMD(DYS-HAC2) clones were selected with the addition of G418 to the culture medium (0.5 mg/ml for mesoangioblasts clones; 0.6 mg/ml for myoblasts clones). HeLa cells were kept in DMEM supplemented with 10% FBS, 1% glutamine and 1% penicillin/streptomycin. HUVECs were cultured in EGM media (Lonza) in 5% $CO_2$/3% $O_2$ at 37°C on 1.5% gelatin-coated plastic.

riDMD myoblasts were cultured in Iscove's modified Dulbecco's medium (IMDM; Sigma) containing 10% FBS, 2 mM glutamine, 0.1 mM 2-mercaptoethanol, 1% non-essential amino acids, human basic fibroblast growth factor (5 ng/ml), HGF (5 ng/ml; Sigma), penicillin (100 IU/ml), streptomycin (100 mg/ml), 0.5 mM oleic and linoleic acids (Sigma), 1.5 mM $Fe^{2+}$ [iron(II) chloride tetrahydrate, Sigma; or Fer-In-Sol, Mead Johnson], 0.12 mM $Fe^{3+}$ [iron(III) nitrate nonahydrate, Sigma; or Ferlixit, Aventis] and 1% insulin/transferrin/selenium (Gibco). Colcemid treatment varied according to purpose (MMCT or FISH) and cell type, as detailed in Appendix Table S3.

### Lentiviral vectors

LOX-TERT-IRESTK (human TERT), LOX-GFP-IRESTK, LOX-CWBmi1 (murine Bmi1, 92.4 and 98% degree of homology with human cDNA and protein, respectively; Alkema et al, 1993; Bhattacharya et al, 2015), IDLV NLS-Cre and MyoD-ER (murine MyoD) lentiviral particles were generated and titrated as previously published (Messina et al, 2010). LOX-TERT-IRESTK, LOX-GFP-IRESTK, LOX-CWBmi1 plasmid maps and sequences are available at www.addgene.org/Didier_Trono/. For IDLV NLS-Cre infection, cells were transduced for 24 h. For all other lentiviruses, cells were transduced overnight. All transductions were performed with polybrene (8 μg/ml).

### Telomeric repeat amplification protocol (TRAP) assay

Detection of telomerase activity was done using TRAPeze® Telomerase Detection Kit (Chemicon) following the manufacturer's instructions. Cells were collected and proteins were extracted using 1× CHAPS lysis buffer incubating each sample at 30°C for 30 min, allowing telomerase (if present and active) to add six bases telomeric repeats onto 3′ end of a synthetic oligonucleotide substrate. Lysates were then amplified using PCR generating a ladder of products with six base increments starting at 50 nucleotides in presence of γ-$^{32}$P-ATP end-labelled primers. PCR products were then run on a non-denaturing polyacrylamide gel, and then, the gel was then directly exposed on X-ray films.

### Immunofluorescence

Immunofluorescence analysis was performed as previously published (Tedesco et al, 2012). Briefly, muscle samples were frozen in liquid nitrogen–cooled isopentane and serial 7-μm sections were cut using a cryostat (Leica). Cells or tissue sections were washed with PBS and fixed with 4% PFA (Sigma) at 4°C for 10 min, permeabilised with 0.2% Triton X-100 (Sigma) and 1% BSA (Sigma) in PBS for 30 min at room temperature (RT). Cells/tissue sections were then blocked with 10% donkey and/or goat serum (Sigma) for 30 min at room temperature to reduce secondary antibody background signal. Tissue sections were incubated overnight at 4°C with primary antibodies (list available in Appendix Table S4; sections were not fixed for dystrophin staining). After incubation, samples were washed with 0.2% Triton X-100, 1% BSA in PBS and then incubated with 488, 546, 594 or 647 fluorochrome-conjugated IgGs (Molecular Probes) together with Hoechst dye for 1 h at room temperature in 0.2% Triton X-100/PBS. After three final washes in PBS, dishes or slides were mounted using mounting medium (Dako) and analysed under fluorescence microscopes (Nikon and Leica). Images were analysed using either ImageJ or Photoshop (CS4 or CC; Adobe).

### Western blot

For detecting Bmi1, proteins were extracted using Laemmli buffer with 2% sodium dodecyl sulphate (SDS), 50 mM Tris–HCl pH 6.8 and 10% glycerol. For dystrophin detection, proteins were extracted using 3% SDS, 0.115 M sucrose and 0.066 M Tris–HCl pH 7.5 buffer. Lysates were then incubated on a running wheel at 4°C for 1 h and then centrifuged at 15,000 $g$ for 15 min. Protein concentrations were determined by BCA protein assay (Pierce) using bovine serum albumin for standard curve; 0.1 mM DTT (dithiothreitol) was added after determination. Before loading, samples were denatured at 90°C for 5 min or 70°C for 10 min. In the case of Bmi1, 50 μg of proteins was loaded and separated on 8% polyacrylamide gel, while

for dystrophin 30–40 μg of proteins was loaded and separated on a 7.5% polyacrylamide gel. Run was monitored by using dual colour Standards (Bio-Rad). Proteins were then transferred using iBlot Gel Transfer device (Invitrogen) according to the manufacturer's instructions, saturated with 5% milk in 0.1% Tween-20 (Sigma)–TBS buffer and hybridised overnight at 4°C with mouse anti-Bmi1 (Millipore) or mouse anti-dystrophin MANDYS106 (Nguyen & Morris, 1993) antibodies. Mouse anti-GAPDH (Sigma) or MyHC (Developmental Studies Hybridoma Bank) was used as normalisers. Membranes were then washed four times (10 min each) with 0.1% Tween-20 (Sigma)–TBS buffer and incubated with HRP-conjugated IgGs (Amersham; 1:10,000 dilution) for 1 h at room temperature, washed four times and finally visualised with the ECL immunoblotting detection system (Amersham). Please refer to Appendix Table S4 for a list of all antibodies used for Western blot.

### PCR and quantitative real-time (qRT) PCR analyses

DNA and RNA were extracted with DNeasy Blood and Tissue kit (QIAGEN) and RNeasy mini kit (QIAGEN), respectively. RNA was reverse-transcribed into double-strand cDNA using ImProm™-II Reverse Transcription System (Promega). Bio-Rad CFX96 thermocycler was used for quantitative real-time analysis, and fold change in gene expression was calculated using $\Delta\Delta C_t$ method (fold change = $2^{-\Delta\Delta C_t}$). The complete list of primer sequences and PCR products is available in Appendix Tables S5 and S7.

### *In vitro* cell proliferation assays

For BrdU incorporation assay, $8 \times 10^4$ human mesoangioblasts were seeded in 3.5-cm dishes to test baseline cell proliferation. The day after plating, 50 μM BrdU (Sigma) was added to the culture medium and cells were kept in the incubator for 1 h. Cells were then washed twice with PBS, fixed with 95% ethanol–5% acetic acid solution for 20 min, incubated for 10 min with 1.5 M HCl and permeabilised for 5 min with 0.2% Triton in PBS. Immunofluorescence staining was performed using Anti-BrdU detection kit (Amersham), and nuclei were stained with Hoechst. BrdU$^+$ and Hoechst$^+$ cells were counted using fluorescence microscopy. Proliferation rate was calculated as the percentage of BrdU$^+$ cells on the total number of nuclei (Hoechst$^+$ cells). BrdU incorporation (1 h pulse) was used also for serum/growth factor dependence and cell contact inhibition assays. For each set of experiments and time points (baseline, days 4, 8 and 12), $8 \times 10^4$ cells were seeded. In the case of serum/growth factor dependence assay, cells were cultured in DMEM plus 2% horse serum, 1% glutamine and 1% penicillin/streptomycin from 24 h after initial seeding. BrdU incorporation was performed at days 4, 8 and 12 in growth factor and serum-free media. In the case of cell contact inhibition assay, cells were allowed to reach confluence and BrdU incorporation rate was tested at days 4, 8 and 12. HeLa cells were used as positive control.

Population doubling curves were calculated as previously described (Cudre-Mauroux *et al*, 2003). Briefly, once cells reached optimal confluence (70–80%), they were detached with trypsin, collected, counted and re-seeded. Growth curves were obtained by calculating the number of population doublings (PD) as a proliferation index. PD = log$N$/log2; $N$ = cells collected/cells initially plated.

### Telomere restriction fragment (TRF) assay

DNA was extracted from cells using the Blood & Cell Culture DNA Mini kit (Qiagen), and TRF assay was performed with TeloTAGGG Telomere Length Assay Kit (Roche) according to the manufacturer's instructions. Briefly, genomic DNA was digested using restriction enzymes HinfI and RsaI, which ensure that telomeric and/or subtelomeric DNA is not cut, while non-telomeric DNA is digested to give rise to low molecular weight fragments. Genomic fragments were separated on a 0.8% agarose gel and transferred by capillarity to a nylon membrane. A digoxigenin (DIG)-labelled probe specific for telomeric repeats was used and membranes were incubated with a DIG-specific antibody covalently coupled to alkaline phosphate. Alkaline phosphatase metabolises CDP-Star, a highly sensitive chemiluminescent substrate that the membrane has been incubated with; as a result, visible signals indicate telomere probe and hence the telomeric restriction fragment (TRF) location on the blot. The average TRF length can be determined by comparing the location of the TRF on the blot relative to a molecular weight.

### Microcell-mediated chromosome transfer

CHO(DYS-HAC), A9(DYS-HAC), riDMD(DYS-HAC2) mesoangioblasts and riDMD(DYS-HAC2) myoblasts hybrids were generated by microcell-mediated chromosome transfer (MMCT) as previously described (Koi *et al*, 1989; Katoh *et al*, 2010; Kazuki *et al*, 2010; Hiratsuka *et al*, 2015). Detailed protocols can be found in Appendix Supplementary Methods.

### DNA construction and HAC engineering

The targeting vector pN for generation of the DYS-HAC2 was constructed in the pBSII backbone vector (Stratagene) using standard ligation technique. The pN contains (i) 3.8- and 2.6-kb fragments for homologous arms corresponding to human chromosome 21 and X locus in AL050305 and AP001657, respectively, (ii) 5′ HPRT gene, (iii) lox71 site, and (iv) CMV promoter-driven neomycin gene flanked by FRT sites. Further details on construction of pN are available upon request. The plasmid pCAG-T7-F encodes the F protein of the Edmonston-B strain under the control of CAG promoter (Katoh *et al*, 2010). The plasmid pTNH6-H-CD13 encodes H protein, the fusion protein of the measles virus and anti-CD13-scFV under the control of CMV promoter in order to re-target the fusogenic activity of the H protein (Hiratsuka *et al*, 2015). The DNA sequence encoding for anti-CD13-scFv was flanked by 5′-SfiI recognition site and 3′-NotI recognition site, synthesised (Blue Heron) and subcloned into pUCminusMCS plasmid (Invitrogen). The SfiI/NotI fragment was subcloned into pTNH6-H at the corresponding restriction sites, resulting in pTNH6-H-CD13. The ORF corresponding to the scFv peptide is in uppercase.

ScFv recognising CD13: gcggcccagccggccGACTACAAGGACGTG GTCATGACTCAGACACCACTGAGCCTGCCCGTTTCCCTCGGCGACC AGGCCAGCATATCATGCCGGAGTTCTCAGTCCATCGTTCACTCAA ACGGCAATACCTATCTGGAATGGTATCTTCAGAAGCCGGGTCAGT CCCCTAAGCTCCTGATCTACAAGGTGTCAAACAGATTTTTCAGGAG TGCCCGACAGATTCTCCGGTAGCGGGTCTGGCACAGATTTCACCC TTAAGATTAGCAGAGTCGAGGCAGAGGACCTGGGCGTTTACTAC TGCTTTCAGGGGTCTCACGTGCCTTGGACCTTTGGTGGGGGTACGA

AACTGGAAATCAAACGCGGCGGGGGGAGGGAGCGGAGGGGGCGGT
TCAGGTGGTGGCGGGTCCGGTGGGGGCGGCAGTCAAGTTCAGCTG
CAACAGTCTGGCGCTGAATTGATGAAACCAGGAGCCAGCGTTAAG
ATCAGCTGTAAGGCGACTGGATACACCTTTAACTCTTACTGGATCG
AATGGGTGAAGCAGCGCCCAGGCCACGGCCTGGAGTGGATCGGAG
AGATCCTCCCAGGCAGCGGGTCTACTAACTACAACGAGAAATTCA
ATGGGAAGGCCACTTTTACAGCAGATGCGTCATCTAACACAGCCT
ATATGCAGCTGTCCTCTCTCACAAGTGAGGATAGCGCCGTGTATT
ATTGTGCGCGCGTGTATTATGGGACATATGGCAGGGTGTATTGGG
GCCAGGGCACGACACTGACTGTCAGTAGTGCTTCTGGCGCAGATg
cggccgcatgatagtaagcggcccagccggcc.

The plasmid pTNH6-Haa-EGFR encodes the fusion protein of the measles H protein and anti-EGFR-scFV under the control of CMV promoter (Nakamura, unpublished). The blasticidin expression plasmid pCMV/Bsd was purchased from Life Technologies.

### Synthesis of p17 and pP-ΔHR plasmids

p17 and pP plasmids were synthesised by Life Technologies after careful in-house designing. Plasmid pP was then modified with FseI digestion and self-ligation with the FseI sites to remove HR sequence (homologous recombination arm for gene insertion in DT40 cells; Hoshiya et al, 2009; Kazuki et al, 2011) in order to shorten pP plasmid and increase efficiency of transfection in target CHO cells.

### DNA transfection

#### DNA transfection for DYS-HAC2 generation

The DYS-HAC1 was previously developed by a cloning of the entire human dystrophin gene into the HAC backbone named 21HAC2 using Cre/loxP-mediated targeted translocation in CHO cells (Hoshiya et al, 2009; Kazuki et al, 2010). DT40 hybrid cells containing the DYS-HAC1 were established by MMCT from CHO hybrids containing the DYS-HAC1 and maintained with 1.5 mg/ml G418 (Invitrogen). The DT40 hybrid cells retaining the DYS-HAC1 were transfected by the electroporation of $1 \times 10^7$ cells with 25 μg of the NotI-linearised pN vector at 25 μF and 550 V in a 4-mm cuvette using a Gene Pulser (Bio-Rad, Hercules, CA, USA). The cells were suspended in basic growth medium and aliquoted into four 96-well flat-bottomed microtiter plates (Becton–Dickinson, Franklin Lakes, NJ, USA). After 2 days, the cells were suspended in selective medium with 1.5 mg/ml G418. About 14 days later, drug-resistant colonies were picked up and expanded for the following analyses.

#### DNA transfection for MMCT using measles virus fusogenic envelope proteins (unconventional MMCT)

$1.5 \times 10^7$ CHO(DYS-HAC2) cells were prepared in 10-cm dishes and then co-transfected with 12 μg of pTNH6-H-CD13 and pCAG-T7-F, respectively, using Lipofectamine 2000 (Invitrogen). 24 h after transfection, transfected CHO(DYS-HAC2) cells were expanded for MMCT experiments.

#### DNA transfection for DYS-HAC3 and DYS-HAC4 generation

$2 \times 10^6$ CHO(DYS-HAC2) cells were co-transfected with p17 (8 μg) and Cre-expressing plasmids (1 μg) using Lipofectamine 2000 (Invitrogen) to induce insertion of p17 into DYS-HAC2 via Cre/LoxP system and generate DYS-HAC3. 24 h after transfection, CHO cells were expanded with a hypoxanthine–aminopterin–thymidine (HAT)-supplemented medium (Sigma-Aldrich) to select only CHO

cells containing DYS-HAC3. CHO(DYS-HAC3) cells were further transfected with pP-ΔHR and Bxb1-expressing plasmid following the same protocol detailed above. Selection was performed according to the following: 8 μg of blasticidin S (Wako), 800 μg of G418 (Promega), 7 mM of L-histidinol dihydrochloride (Sigma-Aldrich) and HAT medium.

### In vitro migration assay

In vitro migration assays were performed using human umbilical vein endothelial cells (HUVECs) as previously published (Giannotta et al, 2014; Bonfanti et al, 2015). Culture inserts (Corning) were incubated for 1 h at 37°C with 1.5% (w/v) gelatin and cross-linked with 2% (w/v) glutaraldehyde solution (Sigma-Aldrich) at room temperature. After 15 min, glutaraldehyde was removed and inserts were incubated with 70% ethanol (v/v) for 1 h, then washed five times with sterile PBS and incubated overnight with PBS containing 2 mM glycine. HUVECs were then plated in 5% $CO_2$/3% $O_2$ at 37°C to reach confluence. Human mesoangioblasts were detached with PBS/1 mM EDTA and labelled with 0.7 μM 6-carboxyfluorescein diacetate (6-CFDA; Molecular Probes, Invitrogen) in suspension for 30 min at 37°C in serum-free media. $10^5$ fluorescent human cells per insert were added to the upper chamber in serum-free MegaCell and were left to migrate in 5% $CO_2$/3% $O_2$ at 37°C for 10 h. After fixing both lower and upper chamber for 10 min with 4% paraformaldehyde, the non-migrated cells on the upper face of the filters were scraped away using a cotton bud, retaining only the cells that had migrated through the HUVEC monolayer. Five pictures for each filter were taken using an inverted fluorescence microscope (Leica), migrated cells on the lower surface were counted and data displayed as numbers of migrated cells per $mm^2$.

### Chromosome analyses (karyotype and fluorescence in situ hybridisation)

Karyotype analyses were performed using either fixed metaphase or interphase spreads stained with quinacrine mustard and Hoechst 33258 to enumerate chromosomes. Images were captured using an AxioImagerZ2 fluorescence microscope (Carl Zeiss GmbH).

For fluorescence in situ hybridisation (FISH), cells were incubated overnight with colcemid (Sigma) to obtain metaphase spreads. Following colcemid treatment, cells were trypsinised, collected, incubated with 0.075 M KCl and washed three times with fresh-made Carnoy's solution (3 parts of methanol, 1 part of acetic acid). Cells were then re-suspended in Carnoy's solution and approximately one drop of cell suspension was spread on each microscope slide [pre-incubated in 50% (v/v) ethanol]. Cells were fixed on microscope slides with a Bunsen's burner and stored for up to a week at −80°C. FISH analyses were performed using either fixed metaphase or interphase spreads of each cell hybrid. Digoxigenin-labelled (Roche) human COT-1 DNA (Invitrogen) and biotin-labelled BAC DNA RP11-954B16 (located in the Dystrophin genomic region, Children's Hospital Oakland Research Institute) were used for the detection of DYS-HAC2 in DT40 (DYS-HAC2), A9(DYS-HAC2)-9 and CHO (DYS-HAC2)-7 cells. Digoxigenin-labelled p11-4 human alpha satellite (centromeres of chromosome 13 and 21, hChr 13/21(cen)) and biotin-labelled RP11-954B16 were used for the detection of DYS-HAC2 in riDMD(DYS-HAC2) clones. Digoxigenin-labelled

human COT-1 DNA and biotin-labelled plasmid pP-ΔHR (immortalising cassette) were used for the detection of DYS-HAC4 in CHO(DYS-HAC4)#19 clone. Digoxigenin-labelled probes were combined with anti-digoxigenin-rhodamine (red)while biotin-labeled probe were combined with anti-biotin-FITC (green). Chromosomal DNA was counterstained with 4′,6-diamidino-2-phenylindole (DAPI, Sigma) or with Giemsa staining, and images were captured using a Leica (Germany) DMI6000B microscope equipped with an AF6000 system and LAS AF 2.3.5 software or AxioImagerZ2 fluorescence microscope (Carl Zeiss GmbH). Additional information is given in Appendix Supplementary Methods.

## Mice and *in vivo* experiments

*Scid/mdx* (C57BL/6 background) and *scid/beige* mice were genotyped as previously described (Tedesco *et al*, 2011, 2012). NSG mice were kindly provided by UCL Biological Services. All mice were kept in specific pathogen-free (SPF) environment, and all the procedures involving living animals are conformed to Italian (D.L.vo 116/92 and subsequent additions) and English law (Animals Scientific Procedure Act 1986 and subsequent additions) and were approved by both San Raffaele Institutional Review Board (IACUC 355) and UK Home Office (Project Licences no. 70/7435 and 70/8566).

For tumorigenesis experiments, both adult (between 2 and 6 months of age) female and male *scid/beige* immunodeficient mice were subcutaneously injected in the dorsal flank with $2 \times 10^6$ cells in 200 μl of PBS (without calcium and magnesium) containing 0.2 international units of sodium heparin (Mayne Pharma). Mice were then regularly examined to detect tumour formation or any signs of distress. Specific follow-up times and the number of mice used are indicated in the respective figure legends or result sections. HeLa cells were used as positive control.

For intramuscular cell transplantation experiments, 2-week-old immunodeficient dystrophic *scid/mdx* and 3-month-old immunodeficient *NOD/scid/gamma* (NSG) male mice were used. In the case of NSG mice, tibialis anterior muscles were subjected to cryoinjury before cell injection as described previously (Meng *et al*, 2015) to promote muscle regeneration. The day before transplantation, MyoD-ER transduced cells were activated with 1 μM 4 OH-tamoxifen. Accordingly, tamoxifen (33 μg/g) was given to mice once per day subcutaneously for three consecutive days starting from 1 day prior to transplantation. Before intramuscular injections, cells were trypsinised, counted, washed/centrifuged twice in PBS and suspended in PBS at the concentration of $5 \times 10^4$ cells/μl. 20 μl of cell suspension (equivalent to $10^6$ cells) was injected in selected hind limb muscles (tibialis anterior for NSG mice; tibialis and gastrocnemius for *scid/mdx* mice). Hind limb muscles from the same mouse were used as contralateral control and injected with 20 μl of PBS. Mice were humanely killed 3 weeks after transplantation and muscles explanted for further analyses. For a detailed protocol, see Gerli *et al* (2014).

For heterotopic subcutaneous transplantation experiments (Sacchetti *et al*, 2016), cells and 3-month-old NSG male mice were pre-treated with tamoxifen as described above. The following day, $10^6$ cells were suspended in 1 ml of cold undiluted reduced growth factor Matrigel (BD cat n 356230; kept on ice) and immediately injected subcutaneously in anesthetised NSG mice with a ice-cold 1 ml syringe (the needle was kept subcutaneously for 1 min to allow

### The paper explained

#### Problem

Duchenne muscular dystrophy (DMD) is a fatal neuromuscular disorder caused by mutations in the dystrophin gene. The large size of the dystrophin gene makes very difficult to clone its full cDNA into conventional delivery vectors for DMD gene therapy.

#### Results

hTERT- and Bmi1-mediated reversible immortalisation of human muscle progenitors allowed their genetic correction with a novel human artificial chromosome (HAC) containing the entire dystrophin locus (DYS-HAC2). Finally, generation of a next-generation multifunctional HAC containing both dystrophin locus and the immortalising genes established a new platform for complex gene transfer.

#### Impact

This study provides the first evidence of translation of HAC technology to clinically relevant human muscle progenitors for *ex vivo* gene therapy of DMD and describes the development of the largest gene therapy vector to date.

solidification of the Matrigel). Mice were humanely killed 2 weeks after injection and Matrigel plugs were explanted and analysed.

## Ethics

Healthy donor mesoangioblasts were obtained from muscle biopsies according to the protocol "Evaluation of regenerative properties of human mesoangioblasts", submitted to the San Raffaele Scientific Institute Ethical Committee by G. Cossu and authorised on 20th February 2007. All patients signed informed consent. DMD human mesoangioblasts were isolated from muscle biopsies kindly provided by Dr. Yvan Torrente (University of Milan, Italy). Human cell work in the Tedesco laboratory was conducted under approval of the NHS Health Research Authority Research Ethics Committee reference no. 13/LO/1826; IRAS project ID no. 141100.

## Statistics

Values were expressed as the mean ± SEM. Significance of the differences between means was assessed by Student's *t*-test or by one-way analysis of variance (ANOVA) followed by Tukey's *post hoc* analyses test when multiple comparisons were required. A probability of less than 5% ($P < 0.05$) was considered to be statistically significant. Data were analysed with GraphPad Prism 6.

**Expanded View** for this article is available online.

## Acknowledgements

We thank D. Trono and M. Cassano for providing immortalising plasmids and immortalised DMD myoblasts; G. Peretti and Y. Torrente for healthy and DMD muscle biopsies, respectively; T. Nakamura for pTNH6-Haa-EGFR and pCAG-T7-F plasmids; J. Chamberlain for MyoD-ER plasmid; M. Amendola for MA1 bidirectional promoter; M. Giannotta for providing HUVECs; J. Meng and J. Morgan for anti-dystrophin antibody; G. Morris and MDA Monoclonal Antibody Resource for providing MANEX1011C and MANDYS106 antibodies. We thank N. Kajitani, C. Sawada, E. Ueno, S. Maffioletti, K. Uno, T. Suematsu, H. Miyamoto

and S. Takata for technical assistance. We are grateful to F. Muntoni, S. Torelli, A. Ferlini and all members of the Tedesco laboratory for helpful discussions and suggestions. This work was supported by Telethon (Project GGP08030), the UK Medical Research Council (Grant No. MR/J006785/1), the European Research Council, and Duchenne Parent Project Onlus. Part of the research leading to these results has received funding from the European Union's Seventh Framework Programme (FP7/2007-2013) under grant agreement no. 602423 (PluriMes) and F5-2009-223098 (OPTISTEM). F.S.T. is funded by a National Institute for Health Research (NIHR) Academic Clinical Fellowship in Paediatrics; this article presents independent research funded by the NIHR and the views expressed are those of the authors and not necessarily those of NHS, NIHR or the Department of Health. F.S.T. is also grateful to the UK Biotechnology and Biological Sciences Research Council (BBSRC), Muscular Dystrophy UK, Duchenne Children's Trust and Duchenne Research Fund for supporting work in his laboratory. The corresponding authors are also grateful to the European Union's Seventh Framework Programme Biodesign Project (Grant No. 262948) and Fundació La Marató de TV3 for supporting their work. Work in M.O. laboratory was supported by JST, CREST, the City Area Program (Basic Stage), JSPS KAKENHI Grant No. 21249022 and Regional Innovation Strategy Support Program from the Ministry of Education, Culture, Sports, Science and Technology of Japan.

## Author contributions

SB performed most of the experimental work with the help of FST, MR and GF, analysed data and wrote the manuscript with FST; HH, NU and YK generated DYS-HAC2 and performed MMCT, FISH and karyotype experiments with KK; RT isolated cells from muscle biopsies; FST conceived DYS-HAC4, HH designed it and engineered it with NU; FST performed transplantation experiments with SB and LAM; AL and LN provided the IDLV NLS-Cre; MM, LP and GD developed the huDYSco; MK generated pTNH6-H-CD13 plasmid; SC and VM performed TRF assay; GM supervised set-up of immortalisation platform and discussed data; MO supervised chromosome engineering and transfer and discussed results; FST and GC planned experiments with SB, discussed data, conceived and coordinated the study and wrote the paper. SB performed this study as a partial fulfilment of her PhD in Morphological and Cytogenetic Science Program at Sapienza University of Rome.

## Conflict of interest

F.S.T. was principal investigator on a research grant to his institution from Takeda New Frontier Science Program, received speaking and consulting fees by Takeda Pharmaceuticals and Sanofi-Genzyme (via UCL Consultants) and has a collaboration with GSK: all for unrelated research projects. The other authors declare that they have no conflict of interest.

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
