## [Review Process File · EMBO Molecular Medicine]

Reversible Immortalisation Enables Genetic Correction of Human Muscle Progenitors and Engineering of Next-Generation Human Artificial Chromosomes for Duchenne Muscular Dystrophy

Sara Benedetti, Narumi Uno, Hidetoshi Hoshiya, Martina Ragazzi, Giulia Ferrari, Yasuhiro Kazuki, Louise Anne Moyle, Rossana Tonlorenzi, Angelo Lombardo, Soraya Chaouch, Vincent Mouly, Marc Moore, Linda Popplewell, Kanako Kazuki, Monotobu Katoh, Luigi Naldini, George Dickson, Graziella Messina, Mitsuo Oshimura, Giulio Cossu and Francesco Saverio Tedesco

Corresponding authors: Giulio Cossu, University of Manchester and Francesco Saverio Tedesco, University College London

Review timeline:

Submission date:	05 November 2016
Editorial Decision:	16 December 2016
Appeal:	22 December 2016
Editorial Decision:	11 January 2017
Revision received:	22 May 2017
Editorial Decision:	30 June 2017
Additional correspondence (author):	14 September 2017
Additional correspondence (author):	28 September 2017
Revision received:	07 November 2017
Accepted:	15 November 2017

Transaction Report:

Editors: Roberto Buccione and Céline Carret

1st Editorial Decision

16 December 2016

Thank you for the submission of your manuscript to EMBO Molecular Medicine. We have now heard back from the Reviewers whom we asked to evaluate your manuscript.

I apologise for the unusual delay in reaching a decision on your manuscript. In this case, we first experienced significant difficulties in securing expert and willing Reviewers. I eventually only managed to secure two reviewers. Further to this the evaluations were delivered with much delay.

I am proceeding based on the two evaluations obtained so far as further delay cannot be justified and would not be productive.

As you will see, while both reviewers fundamentally agree on the technical quality of the study, reviewer 1 finds that the overall advance offered is very limited and lists a number of fundamental concerns to support this contention. They include: 1) the contribution to muscle after transplantation is negligible, 2) unconvincing evidence that HAC-mediated dystrophin expression is occurring and 3) the need to show that DMD-HAC cells can actually be delivered through the vasculature with conversion of fibers to dystrophin positivity. Reviewer 1 also lists a number of other important

concerns. Reviewer 2 appears more positive although s/he also mentions a number of issues including that 1) extended culture appears to degrade myogenic activity and 2) as noted by reviewer 1 as well, the engraftment efficiency is too low.

Our reviewer cross-commenting exercise also highlighted the general concern is that there is excess emphasis on immortalized cells, which does not really represent a substantial advance over previous work describing immortalized myogenic cells and using them for similar studies. Also, the questions whether the HAC strategy can be applied effectively, and whether immortalization can be done efficiently and reversibly remain open. The former question because engraftment is almost negligible. The second because the approach used here is unlikely to be a reflection of what would be used in the clinic.

We agreed that it is extremely unlikely that these concerns could be addressed within a reasonable time frame (3-4 months) and furthermore with no guarantee of success.

Given these fundamental concerns, I have no choice but to return the manuscript to you at this stage. In our assessment it is not realistic to expect to be able to address these issues experimentally and to the satisfaction of the Reviewers in a reasonable time frame.

I am very sorry to have to disappoint you at this stage of analysis. I nevertheless hope that the reviewer comments will be helpful in your continued work in this area.

***** Reviewer's comments *****

Referee #1 (Comments on Novelty/Model System):

Technical aspects are not the weakness of this paper. Novelty is limited by the fact that all of the genetic tools have been described previously. The paper puts them together. As described in my comments, the system would be much more elegant if it were all engineered onto the artificial chromosome.

Referee #1 (Remarks):

The manuscript by Benedetti and colleagues applies several genetic technologies to primary (pericyte-derived mesoangioblast) cells isolated from DMD patients in an effort to generate a cell therapy product that would overcome some of the issues that have dogged cell and gene therapy trials for DMD to date. These include 1. The inability to deliver the full-length dystrophin gene, 2. Problems expanding primary mesoangioblasts, 3. The inability of myoblasts to home to muscle through the circulation. They address these problems by using human artificial chromosome technology, delivering hTERT + BMI1 in floxed lentivectors, and delivering these to pericyte-derived myogenic progenitors, which can enter muscle to some degree, through the circulation, which could in theory enable global delivery.

These are important goals, however there are some significant shortfalls that may or may not be possible to address.

In order of importance:

1. The contribution to muscle after transplantation (Figure 6) is marginal. With injection of one million cells, only a single fiber pair is shown in the SCID/mdx mice, and quantification of such dystrophin-converted fibers is not provided. Perhaps the limited in vivo contribution is related to the extended expansion of these cells? Unless contribution can be improved, it is premature to make claims about therapeutic potential of this system.

2. In the transplantation figure, human Lamin A/C and dystrophin antibodies used the same fluorescent channels. The presumptive dystrophin staining is very weak even though the image is overexposed, so not convincing. It almost looks like it could be low-level incorporation of human Lamin A/C into the cell membrane, which wouldn't normally be visible, but the image is overexposed as is evident when viewing the nuclear Lamin A/C signal. To convince that HAC-mediated dystrophin expression is occurring, it is essential to stain specifically with a dystrophin

antibody in its own independent channel. If the antibodies are not in different species, then don't co-stain - simply stain a serial section with dystrophin only. Quantification of dystrophin+ fibers is also necessary (standard for the field, and not done in this case).

3. The abstract suggests that the expanded cells retain their myogenic potential, however in order to induce myogenesis, the authors need to overexpress MyoD. Since MyoD can be used to induce myogenesis from any cell type, the authors might as well have used skin fibroblasts for this study - they can be isolated less invasively. Thinking about this point, the principal advantage of pericyte-derived cells is that they can be delivered through the vasculature. However, the authors do not take advantage of this aspect of their cells (they deliver by intramuscular injection). So this criticism is really two-fold: 1. can the authors deliver DMD-HAC cells to SCID/mdx mice through the vasculature and see conversion of fibers to dystrophin positivity? 2. Please change the abstract to avoid saying that they retained myogenic potential, but rather indicate that cells were modified with MyoD-ER to enable efficient myogenesis.

4. Issues with the immortalization factors. The authors do not mention their immortalization genes (hTERT and BMI1) in the abstract. BMI1 is an oncogene, actually so is hTERT, so this information is important to highlight. I did not fully understand the negative selection system. The text seems to indicate that ires-HSVTK was only present in the hTERT lentivector. Was it also in the BMI1 vector? If not, isn't it possible that negative selection could miss cells where cre deleted hTERT but missed BMI1? Since the authors did not see tumors form in SCID mice, the lack of ires-HSVTK on BMI1 is not a critical flaw, but the reason for applying negative selection only in the hTERT vector should be discussed.

5. Why don't the authors incorporate the immortalization factors into the HAC? They could be combined with a marker and flanked by LoxP sites allowing later deletion. This would seem to be much more straightforward than independently delivering lentivectors that then need individually to be deleted. Although redoing the whole study with this cleaner approach is probably not feasible for the authors, they may wish to discuss this possibility.

6. To exclude cross-contamination with myogenic progenitors, the authors explain that they negatively select with CD56. However in the results section they do not mention that they also employ positive selection for surface ALP. It would be helpful to include this detail in the results section.

7. The term mesoangioblasts was originally coined to refer to an early embryonic cell with differentiation potential towards endothelial as well as other mesodermal lineages, including muscle. In the current study, the authors use adult pericyte-derived cells and do not show that these have the potency to differentiate into various mesodermal lineages, other than muscle. The use of the term mesoangioblasts for these adult cells has led to their unfortunate conflation with the embryonic cells and confusion over their properties. It would be much more accurate and sincere to refer to these cells as pericyte-derived progenitors or pericyte-derived myogenic progenitors.

8. Is the *Bmi1* gene used in these studies human or murine? If human, please capitalize to make clear. If murine, please indicate in the results section.

Referee #2 (Remarks):

This is a well-written paper exploring solutions for the fact that limited replicative potential in vitro limits our ability to engineer human myogenic cells to be used in therapy. The paper convincingly presents a reversible immortalization strategy that allows extending the replicative potential of primary myogenic cells. The data is high quality and the experiments well controlled. However there are a few conceptual issues to be addressed: The authors introduce the idea of limited in vitro replicative potential as a key limitation affecting the use of primary myogenic cells, and the bulk of in vitro work is done with satellite cell derived myoblasts. However, the in vivo proof of principle was carried out only with mesoangioblasts. Is there a reason for this?

It appears that extended culture still leads to a significant decrease in myogenic activity even in immortalized cells, suggesting that the maintenance of an undifferentiated status (*Pax7* expressing)

may not be supported by Bmi1 and TERT expression. This is a significant limitation of the proposed strategy. Is it the same when using mesangioblasts?

The characterization of the immortalized cells should be extended both during growth and after transgene excision. At a minimum, how many of them are positive for Pax7 MyoD, Myf5 and MyoG? These markers are commonly used to gauge the position of the cells in the myogenic differentiation cascade. What is the spontaneous differentiation rate in culture? Does this change following transgene excision?

Figures 3 g, h strongly suggests that the immortalized cells have a growth advantage. Do cells that retained the immortalizing transgenes following CRE excision have a growth advantage over the successfully excised cells, and will they eventually take over the culture in the absence of counter-selection? Extending the analysis to another time point would reveal this.

The engraftment efficiency seems poor when compared with previously published data from this group. Unfortunately a benchmark is not provided (non-expanded cells?). In general, given that the point of this paper is to propose a solid strategy to generate material for cellular therapy, a quantitative assay showing that this strategy is advantageous over existing approaches would be required.

Appeal

22 December 2016

Following your recent decision letter on the above manuscript, we have been pleased to read the positive assessment of Reviewer #2, but equally disappointed to read some unfair comments by Reviewer #1 which might have negatively influenced your assessment. Please let us summarise in a few points below why we disagree with the outcome of your assessment (followed also by a detailed point-by-point reply to Reviewers' comments):

1. *"The contribution to muscle after transplantation is negligible"*. As discussed in person in Florence, the aim of this paper is not to detail a new pre-clinical cell therapy protocol, but to describe a strategy to overcome the main hurdle limiting HAC transfer into primary human muscle progenitors (leaving to future studies the task of performing detailed functional in vivo experiments based upon this genetic tool). In this context, the cell transplantation experiments were limited to provide proof-of-principle evidence that the product obtained with our approach (i.e. DYS-HAC-corrected DMD myogenic progenitors) were still able to engraft regenerating muscle tissue. Nevertheless, Reviewer #1 ignored what we presented in Figure 6A, where the obtained engraftment is in line with (if not better than) the vast majority of published evidence of muscle xenotransplants (about 150 cells / TA central muscle section with a single injection = 5% of myofibres contained human nuclei). Conversely, Reviewer #1 has selectively decided to limit the evaluation of our in vivo experiments to panels B and C of the same figure, where we only wanted to provide qualitative examples of dystrophin production in mdx/scid mice. Nonetheless, if this is considered critical for the manuscript, we are happy to perform new analysis on the available muscles and new xenotransplants.
2. *"Unconvincing evidence that HAC-mediated dystrophin expression is occurring"*. Unfortunately also in this case it appears that some important experiments were not considered, namely Figures 2D, 2E and 6C clearly show evidence of HAC-mediated dystrophin expression. Also in this case we can easily provide additional evidence of dystrophin expression from the HAC.
3. *"The need to show that DMD-HAC cells can actually be delivered through the vasculature with conversion of fibers to dystrophin positivity"*. As mentioned in point no. 1, it appears that our work has been mistaken as an in-depth pre-clinical study of human mesoangioblast xenotransplantation – something redundant when the same in vivo route of transplant has also been tested even in clinical trials (Cossu G et al., EMBO Mol Med 2015). Experiments centred on vascular delivery would have kept us busy for several months and probably would have not added anything new to the current literature. We believe that it is unfair to shift the focus of the manuscript from the molecular and gene therapy aspects to detailed in

vivo investigations. Moreover, detailed investigations of mice which received intra-arterially delivered DYS-HAC-corrected mouse mesoangioblasts have been already extensively reported (Tedesco FS et al., *Sci Transl Med* 2011, whole figures 5 and 6) – we do not think that it is fully justifiable having to perform the very same set of complex experiments in a much less sensitive set up as (intravascular xenotransplantation is significantly less efficient than the intraspecific one). On the other hand, it could be more informative to perform in vitro assays of endothelial transmigration of DYS-HAC-corrected DMD pericyte-derived mesoangioblasts: this would allow us to test their migration potential using a powerful surrogate assay (human cells on human endothelium using transwell assay, e.g. Giannotta M, Benedetti S et al., *EMBO Mol Med* 2014) which will provide an equally useful outcome without using unnecessary animals. We would be happy to incorporate this new set of experiments in a revised version of the manuscript.

4. “The questions whether the HAC strategy can be applied effectively, and whether immortalization can be done efficiently and reversibly remain open”. Respectfully we do not understand why reversible immortalization is unlikely to benefit clinical development of muscle cell therapies. To date, there is no alternative strategy to overcome the limited proliferation potential of human muscle progenitors besides immortalization or lineage-directed iPSC-differentiation. Notably, clinical translation of the latter is delayed by safety concerns (e.g. tumorigenic potential of undifferentiated cells). Finally, it should be underlined that, despite recent controversial approval of some experimental therapies, clear evidence of significant and long-lasting benefit is still to be seen for any therapeutic strategy for muscular dystrophy.
5. “The system would be much more elegant if it were all engineered onto the artificial chromosome”. As mentioned in the point-by-point reply to Reviewer #1, we have been already working on this project for the past 5 years and have now almost completed the engineering of a novel multifunctional HAC containing: reversibly immortalizing cDNAs, two different suicide genes as safety system, an inducible MyoD cassette (in case of defective myogenesis of target cells) and an additional dystrophin copy to increase gene dosage in target cells (Hoshiya H et. al, unpublished results; Figure R1, below). We planned to detail the cloning and engineering of this new HAC (which we believe will be the largest gene therapy vector ever produced) in a separate study in the near future. However, if you feel that the addition of this new HAC would be essential to the acceptance of the manuscript, we could have a new figure describing this novel HAC within a 4-month timeframe.

Figure R1 – A novel multifunctional HAC for ex vivo gene correction of DMD myogenic progenitors. Simplified scheme of the DYS-HAC16 with functional information enclosed in blue boxes with arrows.

If you would give us the opportunity, we are confident that we can address all Reviewers' points in 4 months. We believe that the resulting manuscript will be significantly improved and may significantly move forward the field of cell therapy for muscular dystrophy where the current therapeutic landscape clearly indicates the need of exploring all possible strategies. We sincerely hope this letter may help you reconsidering your decision.

Point-by-point reply (in blue) to the Reviewers' comments (in grey):

Referee 1 (Comments on Novelty/Model System):

Technical aspects are not the weakness of this paper.

Novelty is limited by the fact that all of the genetic tools have been described previously. The paper puts them together. As described in my comments, the system would be much more elegant if it were all engineered onto the artificial chromosome.

Thanks for the positive technical assessment of our work. We respectfully disagree that all genetic tools have been described, as the manuscript starts with a whole figure detailing a novel HAC (Figure 1). Reviewer #1 thinks that the work would be more elegant if all the immortalizing transgenes were all engineered onto the HAC. Indeed we have been working on this very project for the past 5 years (funded by an MRC translational stem cell grant and by Duchenne Parent Project) and have now almost completed the engineering of a novel multifunctional HAC containing: reversibly immortalizing cDNAs, two different suicide genes as safety system, an inducible MyoD cassette (in case of defective myogenesis of target cells) and an additional dystrophin copy to increase gene dosage in target cells (Hoshiya H et al., unpublished results). A figure with preliminary results has been provided to the Editor and is available upon request (Figure R1, confidential). We will certainly mention this new strategy in the revised version of the Discussion.

Referee #1 (Remarks):

The manuscript by Benedetti and colleagues applies several genetic technologies to primary (pericyte-derived mesoangioblast) cells isolated from DMD patients in an effort to generate a cell therapy product that would overcome some of the issues that have dogged cell and gene therapy trials for DMD to date. These include 1. The inability to deliver the full-length dystrophin gene, 2. Problems expanding primary mesoangioblasts, 3. The inability of myoblasts to home to muscle through the circulation. They address these problems by using human artificial chromosome technology, delivering hTERT + BMI1 in floxed lentivectors, and delivering these to pericyte-derived myogenic progenitors, which can enter muscle to some degree, through the circulation, which could in theory enable global delivery.

We thank Reviewer #1 for the evaluation of our manuscript. However, the described genetic technologies have been applied also to human myoblasts (Figure 2) and not only to mesoangioblasts.

These are important goals, however there are some significant shortfalls that may or may not be possible to address.

In order of importance:

1. The contribution to muscle after transplantation (Figure 6) is marginal. With injection of one million cells, only a single fiber pair is shown in the SCID/mdx mice, and quantification of such dystrophin-converted fibers is not provided. Perhaps the limited in vivo contribution is related to the extended expansion of these cells? Unless contribution can be improved, it is premature to make claims about therapeutic potential of this system.

We thank Reviewer #1 for acknowledging the importance of the goals of our work. We believe that some of the shortfalls currently concerning this Reviewer are the result of possible misinterpretation of our aims and experimental work. Notably, we are confident that they can be clarified or addressed within a reasonable timeframe.

Cell transplantation experiments were designed to provide proof-of-principle evidence that the product obtained with our approach (i.e. DYS-HAC-corrected DMD myogenic progenitors) was still able to engraft regenerating muscle tissue. This was indeed the case as shown in Figure 6A, where the obtained engraftment is in line with (if not better than) the vast majority of published evidence of muscle xenotransplants (about 150 cells / TA central muscle section with a single injection = 5% of myofibres contained human nuclei). We apologise for not having provided additional evidence of dystrophin production in this panel (6A), which was due to the notoriously challenging staining using the human-specific DYS3 antibody in NSG mice (i.e. mouse-dystrophin-positive). We can certainly address this issue with molecular analysis on those very same muscle sections. However, we believe that it is unfair to selectively limit the evaluation of our in vivo experiments to panels B and C of the same figure, where we only wanted to provide some

qualitative examples of dystrophin production in mdx/scid mice (which are much less permissive than NSG mice for xenotransplants). Nonetheless, we are happy to extend the *in vivo* section performing new analysis on the available muscles and new xenotransplants.

2. In the transplantation figure, human Lamin A/C and dystrophin antibodies used the same fluorescent channels. The presumptive dystrophin staining is very weak even though the image is overexposed, so not convincing. It almost looks like it could be low-level incorporation of human Lamin A/C into the cell membrane, which wouldn't normally be visible, but the image is overexposed as is evident when viewing the nuclear Lamin A/C signal. To convince that HAC-mediated dystrophin expression is occurring, it is essential to stain specifically with a dystrophin antibody in its own independent channel. If the antibodies are not in different species, then don't co-stain - simply stain a serial section with dystrophin only. Quantification of dystrophin+ fibers is also necessary (standard for the field, and not done in this case).

We are surprised to see this technical point listed as the second most important concerns, as Reviewer #1 started their assessment with a statement that technical aspects were not the weakness of this paper. Unfortunately also in this case Reviewer #1 appears to have ignored Figure 6A and focused their assessment to a qualitative example of a proof-of-principle transplant. The use of two antibodies with clear different subcellular localization is widely accepted in the field (e.g. Rozkalne A et al., *Hum Gen Ther* 2014) and the conclusion that what we have observed is low-level incorporation of Lamin A/C in the cell membrane (which is biologically impossible, as it's a nuclear envelope protein) and that our image is overexposed is not supported by evidence. We would have co-stained serial sections, but this would have not enabled visualisation of double-positive fibres (as nuclear thickness is smaller than the 7micron section thickness). As discussed in the point above, we apologise for not having quantified the number of dystrophin-positive fibres and we will include this information in the revised version of the manuscript.

3. The abstract suggests that the expanded cells retain their myogenic potential, however in order to induce myogenesis, the authors need to overexpress MyoD. Since MyoD can be used to induce myogenesis from any cell type, the authors might as well have used skin fibroblasts for this study - they can be isolated less invasively. Thinking about this point, the principal advantage of pericyte-derived cells is that they can be delivered through the vasculature. However, the authors do not take advantage of this aspect of their cells (they deliver by intramuscular injection). So this criticism is really two-fold: 1. can the authors deliver DMD-HAC cells to SCID/mdx mice through the vasculature and see conversion of fibers to dystrophin positivity? 2. Please change the abstract to avoid saying that they retained myogenic potential, but rather indicate that cells were modified with MyoD-ER to enable efficient myogenesis.

We respectfully disagree with Reviewer #1 on this point. Indeed we have clearly and repeatedly shown that both myoblasts and mesoangioblasts undergo spontaneous myogenic differentiation after reversible immortalization and HAC transfer (Figures 2E, 3J and supplementary figure 3F). We have also provided examples of enhanced myogenesis upon MyoD-ER expression, to show that cells can achieve very high levels of myogenic differentiation using this genetic tool (>90%). This is an important point, as any myogenic cells could potentially decrease its differentiation potential during high density cultures necessary for HAC transfer and this strategy could be utilised as an efficacious contingency plan. For this reason we decided to perform intramuscular transplants with cells responsive to MyoD induction. Indeed this approach was used to show re-establishment of myogenic potential in mesoangioblasts that lost it (Morosetti R et al., *PNAS* 2006). Moreover, short and controlled MyoD expression in human iPSC-derived mesoangioblast-like cells has been showed not to interfere with their migration, engraftment and differentiation potential (Tedesco FS et al., *Sci Transl Med* 2012). On this point, the statement that all cells respond equally to MyoD-mediated myogenesis is incorrect, as it has been demonstrated that the process is cell type dependent and that efficiency increases with lineage proximity to skeletal muscle (Davis RL et al., *Cell* 1987 [please see table 1]; Schafer BW et al., *Nature* 1990). In view of the above reasons, we believe that the abstract does not need major changes, although we will mention the possibility of enhancing myogenesis in the same cells by means of MyoD-ER.

Regarding intravascular delivery, it appears that our work has been mistaken as an *in depth* pre-clinical *in vivo* study on human mesoangioblast xenotransplantation – something redundant when the same *in vivo* route of transplant has also been tested even in clinical trials (Cossu et al., *EMBO Mol Med* 2015). We believe that it is unfair to completely shift the focus of the manuscript

from the molecular and gene therapy aspects to detailed and complex *in vivo* investigations (which apply to any form of genetic correction, not necessary HACs). Moreover, detailed investigations of mice which received intra-arterially delivered DYS-HAC-corrected mouse mesoangioblasts have been already extensively reported (Tedesco FS et al., *Sci Transl Med* 2011, whole figures 5 and 6) and we do not think that it is fully justifiable having to perform the very same set of complex experiments in a much less sensitive set up as (intravascular xenotransplantation is significantly less efficient than the intraspecific one). On the other hand it could be much more informative to perform *in vitro* assays of endothelial transmigration of DYS-HAC corrected DMD pericyte-derived mesoangioblasts: this would allow us to test their migration potential using a powerful surrogate assay (human cells on human endothelial cells using transwell assay, e.g. Giannotta, Benedetti et al., *EMBO Mol Med* 2014) which will provide an equally useful outcome without using unnecessary animals. We would be more than happy to incorporate this new set of exciting experiments in a revised version of the manuscript.

4. Issues with the immortalization factors. The authors do not mention their immortalization genes (hTERT and BMI1) in the abstract. BMI1 is an oncogene, actually so is hTERT, so this information is important to highlight. I did not fully understand the negative selection system. The text seems to indicate that ires-HSVTK was only present in the hTERT lentivector. Was it also in the BMI1 vector? If not, isn't it possible that negative selection could miss cells where cre deleted hTERT but missed BMI1? Since the authors did not see tumors form in SCID mice, the lack of ires-HSVTK on BMI1 is not a critical flaw, but the reason for applying negative selection only in the hTERT vector should be discussed.

We thank Reviewer #1 for the suggestion of mentioning the specific genes in the abstract – we will certainly do so. We apologise for not having provided additional information on the negative selection system. The ires-HSVTK cassette is present in the hTERT vector but not on the Bmi1 one for cloning space reasons and to reduce toxicity. This was made possible by the fact that myoblasts containing only one of the two transgenes do not immortalise and hence cells potentially harbouring only the Bmi1 cassette would be lost during amplification (Cudre-Mauroux C et al., *Hum Gen Ther* 2003). Moreover, on top of the tumorigenesis assays correctly quoted by this Reviewer, we have also provided evidence that ganciclovir selection did not miss relevant cells, as the Bmi1 cassette was undetectable by PCR analysis (Figure 4E). Nonetheless, we will certainly provide additional details and discuss this specific point in the revised version of our manuscript.

5. Why don't the authors incorporate the immortalization factors into the HAC? They could be combined with a marker and flanked by LoxP sites allowing later deletion. This would seem to be much more straightforward than independently delivering lentivectors that then need individually to be deleted. Although redoing the whole study with this cleaner approach is probably not feasible for the authors, they may wish to discuss this possibility.

We thank again Reviewer #1 for this interesting comment, which we have already discussed above in response to their first comment.

6. To exclude cross-contamination with myogenic progenitors, the authors explain that they negatively select with CD56. However in the results section they do not mention that they also employ positive selection for surface ALP. It would be helpful to include this detail in the results section.

We apologise with Reviewer #1 for not having specified this point in the manuscript. This will surely be done in its revised version.

7. The term mesoangioblasts was originally coined to refer to an early embryonic cell with differentiation potential towards endothelial as well as other mesodermal lineages, including muscle. In the current study, the authors use adult pericyte-derived cells and do not show that these have the potency to differentiate into various mesodermal lineages, other than muscle. The use of the term mesoangioblasts for these adult cells has led to their unfortunate conflation with the embryonic cells and confusion over their properties. It would be much more accurate and sincere to refer to these cells as pericyte-derived progenitors or pericyte-derived myogenic progenitors.

We understand this Reviewer's point and unfortunately this is a recurrent issue. After more than 50

papers published on the topic, it is now difficult to correct this initial improper definition. Mesoangioblasts are defined as the *in vitro* expanded counterpart of cells from the vessel wall. This might be vague as it is, for example, the definition of “mesenchymal stem cells”. What we can do is to add in the footnotes that post-natal mesoangioblasts are the *in vitro* expanded counterpart of a subset of pericytes whereas embryonic mesoangioblasts are a subset of endothelial cells. Nonetheless, if deemed necessary we are also happy to amend the terminology to pericyte-derived myogenic progenitors.

8. Is the *Bmi1* gene used in these studies human or murine? If human, please capitalize to make clear. If murine, please indicate in the results section.

We apologise for not having specified this information in the original manuscript and thank the Reviewer for highlighting it. The *Bmi1* cDNA used for this study is murine (due to the very high homology and readily availability) and we will acknowledge it in the appropriate section of the revised manuscript.

Referee #2 (Remarks):

This is a well-written paper exploring solutions for the fact that limited replicative potential *in vitro* limits our ability to engineer human myogenic cells to be used in therapy. The paper convincingly presents a reversible immortalization strategy that allows extending the replicative potential of primary myogenic cells. The data is high quality and the experiments well controlled. However there are a few conceptual issues to be addressed: The authors introduce the idea of limited *in vitro* replicative potential as a key limitation affecting the use of primary myogenic cells, and the bulk of *in vitro* work is done with satellite cell derived myoblasts. However, the *in vivo* proof of principle was carried out only with mesangioblasts. Is there a reason for this?

We thank Reviewer #2 for the very positive assessment of our manuscript. We are glad to see that they have rated our paper as “well-written”, the reversible immortalization strategy as “convincing”, our data as “high quality” and the experiments as “well controlled”. We apologise if this was not highlighted in the manuscript, but actually only initial experiments (e.g. Figure 2) were performed with myoblasts and the bulk of the work was actually then performed with pericyte-derived mesoangioblasts (e.g. Figures 3,4,5 and 6). We started from myoblasts as a proof-of-principle set of experiments given their availability (Cudre-Mauroux C et al., Hum Gen Ther 2003) and then we moved to pericytes for *in vivo* experiments. If deemed essential, we could perform *in vivo* experiments also with human myoblasts and incorporate them in the revised version of the manuscript.

It appears that extended culture still leads to a significant decrease in myogenic activity even in immortalized cells, suggesting that the maintenance of an undifferentiated status (*Pax7* expressing) may not be supported by *Bmi1* and TERT expression. This is a significant limitation of the proposed strategy. Is it the same when using mesangioblasts?

We thank Reviewer #2 for raising this interesting discussion point, which will be certainly incorporated in the revised discussion section of our manuscript. We do not think that reversible immortalization directly interferes with the maintenance of a *Pax7*-positive undifferentiated status, as *Bmi1* has even been shown to play a key role in maintaining satellite cell pool in postnatal skeletal muscle (with reduced *Pax7*⁺/*Myf5*⁻ satellite cells found in *Bmi1*-null mice; Robson LG et al., Plos One 2011). What is likely to happen is the immortalization and subsequent expansion of cells that might have already lost *Pax7* expression during the initial expansion prior to lentiviral transduction, hence the following amplification is likely to expand committed progenitors. Recent advances in human satellite cell purification (e.g. Xu X et al., Stem Cell Reports 2015) together with immortalization at very early passages (e.g. P1-2) could improve this outcome. The same process could be less relevant for pericyte-derived mesoangioblasts, whose dependence from *Pax7* is less clear (Dellavalle A et al., Nat Cell Biol 2007).

Regarding the decrease in myogenic activity, this could also be a consequence of high-density cultures during MMCT (the technique necessary to perform HAC transfer). In the event that this decrease could impact on engraftment and differentiation *in vivo*, a contingency plan with MyoD-ER-induced rescue of differentiation has been described in the manuscript (figures 3J and 6). We would be delighted to discuss the above points in the revised Discussion section of our

manuscript.

The characterization of the immortalized cells should be extended both during growth and after transgene excision. At a minimum, how many of them are positive for Pax7 MyoD, Myf5 and MyoG? These markers are commonly used to gauge the position of the cells in the myogenic differentiation cascade. What is the spontaneous differentiation rate in culture? Does this change following transgene excision?

We apologise with Reviewer #2 for not having performed the above-mentioned experiments for the current manuscript version. Unfortunately the high-volume of genetic engineering experiments for HAC generation and transfer has partially shifted the focus from those interesting cell biology aspects. We will now address them in the revised version of the manuscript.

Figures 3 g, h strongly suggests that the immortalized cells have a growth advantage. Do cells that retained the immortalizing transgenes following CRE excision have a growth advantage over the successfully excised cells, and will they eventually take over the culture in the absence of counter-selection? Extending the analysis to another time point would reveal this.

We thank Reviewer #2 for raising also this interesting discussion point. It is indeed likely that cells that might fail to excise the immortalizing cassettes following Cre administration could have a growth advantage in the absence of counter-selection. Indeed this is why Ganciclovir has been administered within two weeks from CRE-mediated transgene excision (Figure 4D,E). Please accept our apologies, but we did not understand which analysis this Reviewer wanted to extend to another time point.

The engraftment efficiency seems poor when compared with previously published data from this group. Unfortunately a benchmark is not provided (non-expanded cells?). In general, given that the point of this paper is to propose a solid strategy to generate material for cellular therapy, a quantitative assay showing that this strategy is advantageous over existing approaches would be required.

The aim of our manuscript is not to detail a new pre-clinical cell therapy protocol, but to describe a strategy to overcome the main hurdle limiting HAC transfer into primary human muscle progenitors (leaving to future studies the task of performing detailed functional in vivo experiments based upon this genetic tool). In this context, the cell transplantation experiments were limited to provide proof-of-principle evidence that the product obtained with our approach (i.e. DYS-HAC-corrected DMD myogenic progenitors) were still able to engraft regenerating muscle tissue. Nevertheless, what we presented in Figure 6A is actually in line with (if not better than) the vast majority of published evidence of muscle xenotransplants (about 150 cells / TA central muscle section with a single injection = 5% of myofibres contained human nuclei). We agree with Reviewer #2 that previous data from our groups have reported better engraftment figures, but those were based upon intraspecific (i.e. mouse cells into mouse muscles; e.g. Tedesco FS et al., *Sci Transl Med* 2011), a procedure notoriously more efficient than human xenotransplants in mouse muscles (even in immunodeficient animals, due to their well-functioning innate immunity). Regarding the possibility to benchmark this with existing approaches, we are not sure that this experiment would be informative, as this is not an alternative to muscle stem cell transplant or a better way to do so, but it is rather a strategy to overcome the impossibility to do it after HAC transfer. Nonetheless, we propose to expand the engraftment analysis by better quantifying donor-derived human dystrophin fibres and insert a meta-analysis table cross comparing our figures with published xenotransplantation data from our and other groups.

2nd Editorial Decision

11 January 2017

Thank you for your message asking to reconsider our recent decision on your manuscript entitled "Reversible Immortalization Enables Human Artificial Chromosome-Mediated Genetic Correction of Transplantable Human Dystrophic Muscle Progenitors".

I apologize for the unusual delay in providing you with an answer. We are currently dealing with a tremendous backlog of manuscripts, due also to the holiday season. Furthermore, new submissions

must take some precedence over appeals.

We have now re-discussed your manuscript and point-by-point rebuttal and have consequently decided to allow you to prepare and submit a substantially revised manuscript, with the understanding that the Reviewers' concerns must be addressed with additional experimental data where appropriate and that acceptance of the manuscript will entail a second round of review. We do agree however that concerning point 3 of your rebuttal, it would be sufficient to test the migration potential of corrected mesoangioblasts in the Transwell assay, and concerning point 1, a rebuttal limited to discussion might also be sufficient. Finally, regarding the novel multifunctional mega-HAC the reviewer asked about: we would ideally like to see this too as it would increase the overall appeal of the manuscript. However, we also appreciate that you wish to publish separately. I will leave it up to you, but if you decide not to include the new HAC, it will not be a basis for rejection.

Please note that it is EMBO Molecular Medicine policy to allow a single round of revision only and that, therefore, acceptance or rejection of the manuscript will depend on the completeness of your responses included in the next, final version of the manuscript.

I look forward to seeing a revised form of your manuscript as soon as possible.

1st Revision - authors' response

22 May 2017

Referee 1

Technical aspects are not the weakness of this paper. Novelty is limited by the fact that all of the genetic tools have been described previously. The paper puts them together.

We thank Reviewer #1 for the positive technical assessment of our work, but we respectfully disagree that all genetic tools have been already described, as the manuscript starts with a whole figure detailing a novel HAC, namely DYS-HAC2 (Fig 1). Moreover, as described in more detail below, we have now added to the revised version of the manuscript a novel multifunctional DYS-HAC (new Fig 7 and Appendix Fig S3 and S4), which is the largest and possibly most complex gene therapy vector developed to date.

As described in my comments, the system would be much more elegant if it were all engineered onto the artificial chromosome.

We thank Reviewer #1 for this comment. Indeed we have been working for several years on the generation of a novel HAC that, amongst various other functions, could also deliver genomic-integration-free genetic correction and reversible immortalisation. Although we were planning to publish this story as a separate paper in the future, following Reviewer's #1 comment, we have now decided to include it in the revised version of this manuscript. Indeed there are now three new figures (Fig 7 and Appendix Figure S3 and S4) describing the generation of the aforementioned multifunctional HAC, namely DYS-HAC4. DYS-HAC4 contains: 1) the Dystrophin locus (2.4Mb) for complete genetic correction; 2) a hTERT and Bmi1 immortalizing cassette with an elimination system via CreERT2/loxP system and a negative selection strategy by Ganciclovir-TK; 3) a clinically-tested safeguard system based upon inducible Caspase 9 (iCasp9) which can induce cell apoptosis following treatment with the drug AP1903; 4) an inducible skeletal myogenic differentiation system based upon MYOD-ER; 5) a codon-optimised human dystrophin cDNA (huDYSco) to boost dystrophin expression on top of the endogenous locus. The new Figures describe DYS-HAC4 design (Fig 7A and Appendix Fig S3), successful cloning of novel genes onto DYS-HAC backbone (Fig 7B, C and D), presence of single episomal copy of DYS-HAC4 in donor cells (Fig 7E), and expression of novel genes (Fig 7F). Transfer of this new HAC into target human cells and transplanting them into specific animal models will require several more months of work and will be the subject of a future paper. Nonetheless, we believe that the addition of this whole new section substantially improves the quality of the current manuscript and opens new avenues for gene therapy, to the point that this is now reflected by our new title (*Reversible Immortalisation Enables Genetic Correction of Human Muscle Progenitors and Engineering of Next-Generation Human Artificial Chromosomes for Duchenne Muscular Dystrophy*).

The manuscript by Benedetti and colleagues applies several genetic technologies to primary

(pericyte-derived mesoangioblast) cells isolated from DMD patients in an effort to generate a cell therapy product that would overcome some of the issues that have dogged cell and gene therapy trials for DMD to date. These include 1. The inability to deliver the full-length dystrophin gene, 2. Problems expanding primary mesoangioblasts, 3. The inability of myoblasts to home to muscle through the circulation. They address these problems by using human artificial chromosome technology, delivering hTERT + BM11 in floxed lentivectors, and delivering these to pericyte-derived myogenic progenitors, which can enter muscle to some degree, through the circulation, which could in theory enable global delivery.

These are important goals, however there are some significant shortfalls that may or may not be possible to address.

In order of importance:

1. The contribution to muscle after transplantation (Figure 6) is marginal. With injection of one million cells, only a single fibre pair is shown in the SCID/mdx mice, and quantification of such dystrophin-converted fibres is not provided. Perhaps the limited in vivo contribution is related to the extended expansion of these cells? Unless contribution can be improved, it is premature to make claims about therapeutic potential of this system.

We thank Reviewer #1 for acknowledging the importance of our goals. We believe that some of the concerns above may be the result of possible misinterpretation of our aims. Cell transplantation experiments were designed to provide proof-of-principle evidence that the product obtained with our approach (i.e. DYS-HAC-corrected DMD myogenic progenitors) was still able to engraft regenerating muscle tissue. This was indeed the case, as shown in Figure 6B, where the obtained engraftment is in line with the vast majority of published evidence of muscle xenotransplants (about 150 cells / TA central muscle section with a single injection = 5% of myofibres contained human nuclei). However, we believe that it might be unfair to selectively limit the evaluation of our in vivo experiments to panels B and C of the same figure, where we only wanted to provide some qualitative examples of dystrophin production in mdx/scid mice (which are much less permissive than NSG mice for xenotransplants). Nonetheless to address this Reviewer's concern we have provided new data on transplantation of HAC-corrected human myoblasts (Fig. 6A) and focused the manuscript more on the technological aspects of HAC engineering for gene therapy (e.g. new mega-HAC, Figure 7 and Appendix Figures S2 and S3), toning down the in vivo aspect of the work (e.g. we have removed the word "transplantable" from the title).

2. In the transplantation figure, human Lamin A/C and dystrophin antibodies used the same fluorescent channels. The presumptive dystrophin staining is very weak even though the image is overexposed, so not convincing. It almost looks like it could be low-level incorporation of human Lamin A/C into the cell membrane, which wouldn't normally be visible, but the image is overexposed as is evident when viewing the nuclear Lamin A/C signal. To convince that HAC-mediated dystrophin expression is occurring, it is essential to stain specifically with a dystrophin antibody in its own independent channel. If the antibodies are not in different species, then don't co-stain - simply stain a serial section with dystrophin only. Quantification of dystrophin+ fibres is also necessary (standard for the field, and not done in this case).

We were surprised to see this technical point listed as the second most important concern, as Reviewer #1 started their assessment with a statement that technical aspects were not the weakness of this paper. The use of two antibodies with different subcellular localization is widely accepted in the field (e.g. Rozkalne et al., *Hum Gen Ther* 2014) and the conclusion that what we have observed is low-level incorporation of Lamin A/C in the cell membrane is not supported by evidence and difficult to explain biologically. We could have co-stained serial sections, but this would have not enabled direct visualisation of double-positive fibres but only indirect evidence that the human nucleus was inside a dystrophin positive fibre. Nonetheless, we have now provided additional molecular evidence of human dystrophin expression in vivo (Fig. 6A) and shifted the focus of the study from transplantation (which was only aimed at providing a proof-of-principle example) to vector engineering (Fig. 7).

3. The abstract suggests that the expanded cells retain their myogenic potential, however in order to induce myogenesis, the authors need to overexpress MyoD. Since MyoD can be used to induce myogenesis from any cell type, the authors might as well have used skin fibroblasts for this study - they can be isolated less invasively. Thinking about this point, the principal advantage of pericyte-derived cells is that they can be delivered through the vasculature. However, the authors do not take

advantage of this aspect of their cells (they deliver by intramuscular injection). So this criticism is really two-fold: 1. can the authors deliver DMD-HAC cells to SCID/mdx mice through the vasculature and see conversion of fibres to dystrophin positivity?

2. Please change the abstract to avoid saying that they retained myogenic potential, but rather indicate that cells were modified with MyoD-ER to enable efficient myogenesis.

We respectfully disagree with Reviewer #1 on this point. Indeed we have shown that both myoblasts and mesoangioblasts undergo spontaneous myogenic differentiation after reversible immortalisation and HAC transfer (Fig 2D-F, Fig 3J and Fig EV1B, C and Fig EV2F). We have also provided examples of enhanced myogenesis upon MyoD-ER expression, to show that cells can achieve higher levels of myogenic differentiation using this genetic tool (Fig 3J and Fig EV2F). This is an important point, as any myogenic cells could decrease its differentiation potential during high density cultures necessary for HAC transfer and this strategy could be utilised as an efficacious contingency plan. Indeed this approach was used to show re-establishment of myogenic potential in human mesoangioblasts that lost it (Morosetti R et al., *PNAS* 2006). For this reason we decided to perform intramuscular transplants with cells responsive to MyoD induction. Moreover, short and controlled MyoD expression in human iPSC-derived mesoangioblast-like cells has been showed not to interfere with their migration, engraftment and differentiation potential (Tedesco FS et al., *Sci Transl Med* 2012).

The statement that all cells respond equally to MyoD-mediated myogenesis is questionable, as it has been demonstrated that the process is cell type dependent and that efficiency increases with lineage proximity to skeletal muscle and chromatin status (Davis RL et al., *Cell* 1987 [please see table 1]; Schafer BW et al., *Nature* 1990; Albini S et al., *Cell Rep* 2013). Nonetheless, we agree with this Reviewer that more details should be available in the abstract regarding the use of MyoD and we have now added this information (“Cells remained myogenic in vitro (spontaneously or upon MyoD induction)”).

As for intravascular delivery, it appears that our work has been considered an in depth pre-clinical in vivo study on human mesoangioblast xenotransplantation – something possibly redundant when the same in vivo route of transplantation has also been tested in clinical trials (Cossu et al., *EMBO Mol Med* 2015). We believe that the focus of the manuscript is and should be evaluated as the generation of a novel gene therapy strategy. Moreover, detailed studies of mice which received intra-arterially delivered DYS-HAC-corrected mouse mesoangioblasts have been already extensively reported (Tedesco FS et al., *Sci Transl Med* 2011, whole figures 5 and 6) and we do not think that it is fully justifiable having to perform the same set of complex experiments in a less sensitive set up (as intravascular xenotransplantation is less efficient than the intraspecific route). On the other hand, it could be much more informative to perform in vitro assays of endothelial transmigration of DYS-HAC corrected DMD pericyte-derived mesoangioblasts: this would allow us to test their migration potential using a powerful surrogate assay (human cells on human endothelial cells using transwell assay, e.g. Giannotta & Benedetti et al., *EMBO Mol Med* 2014; Bonfanti et al, 2015 *Nat Comm*) which will provide an equally useful outcome without using unnecessary animals. We have now performed this set of experiments and incorporated the results in Fig 5F. We showed that riDMD(DYS-HAC2) mesoangioblasts clones and riDMD mesoangioblasts transmigrate with the same extent of human healthy and not immortalised mesoangioblasts (50.7 ± 6.2 migrated cells/mm²; 55.9 ± 9.3 migrated cells/mm²; 44.3 ± 6.9 migrated cells/mm²; Fig 5F and Fig EV3D). Moreover, all human mesoangioblasts tested here, regardless of immortalisation or genetic correction, transmigrate significantly more than human myoblasts (15 ± 1.4 migrated cells/mm²; Fig 5F). This data shows that immortalisation and DYS-HAC transfer do not interfere with human mesoangioblasts intrinsic transmigration potential.

4. Issues with the immortalisation factors. The authors do not mention their immortalisation genes (hTERT and BM1) in the abstract. BM1 is an oncogene, actually so is hTERT, so this information is important to highlight. I did not fully understand the negative selection system. The text seems to indicate that ires-HSVTK was only present in the hTERT lentivector. Was it also in the BM1 vector? If not, isn't it possible that negative selection could miss cells where cre deleted hTERT but missed BM1? Since the authors did not see tumors form in SCID mice, the lack of ires-HSVTK on BM1 is not a critical flaw, but the reason for applying negative selection only in the hTERT vector should be discussed.

We thank Reviewer #1 for this comment, which has also prompted us to perform an interesting experiment. We have now mentioned the specific genes in the abstract and we apologise for not

having provided additional information on the negative selection system. The HSV1-TK cDNA was transcriptionally linked only with hTERT using an IRES sequence; this reduced transgene size and the likelihood of a bystander effect elicited by few TK-expressing cells inducing cell death of TK-negative neighbouring cells (Freeman SM et al., 1993 *Cancer Res*; Denning & Pitts JD *Hum Gene Ther* 1997). Additionally, this does not represent a problem for our strategy, as myogenic cells containing only one of the two transgenes (either Bmi1 or hTERT) do not immortalise and hence cells potentially harbouring only the Bmi1 cassette would be lost during amplification (Cudré-Mauroux C et al., *Hum Gene Ther* 2003). We have now provided these additional details and discuss this specific point in the revised version of our manuscript (page 13). Moreover, on top of the tumorigenesis assays correctly quoted by this Reviewer, in the first version of our manuscript we had already provided evidence that ganciclovir selection did not miss cells that might have escaped Cre-mediated transgene-excision, as the Bmi1 transgene was undetectable by PCR analysis following negative selection (Fig 4E).

5. Why don't the authors incorporate the immortalisation factors into the HAC? They could be combined with a marker and flanked by LoxP sites allowing later deletion. This would seem to be much more straightforward than independently delivering lentivectors that then need individually to be deleted. Although redoing the whole study with this cleaner approach is probably not feasible for the authors, they may wish to discuss this possibility.

We thank again Reviewer #1 for this very interesting comment. As discussed above in reply to their first comment, we have now added to the revised manuscript the engineering of the multifunctional DYS-HAC4 containing a floxed immortalisation cassette among other novel genes (Fig 7 and Appendix Fig S3 and S4).

6. To exclude cross-contamination with myogenic progenitors, the authors explain that they negatively select with CD56. However in the results section they do not mention that they also employ positive selection for surface ALP. It would be helpful to include this detail in the results section.

We apologise with Reviewer #1 for not having specified this point in the manuscript. We have now added this information in the Results (page 8) and Materials and Methods (page 23) sections.

7. The term mesoangioblasts was originally coined to refer to an early embryonic cell with differentiation potential towards endothelial as well as other mesodermal lineages, including muscle. In the current study, the authors use adult pericyte-derived cells and do not show that these have the potency to differentiate into various mesodermal lineages, other than muscle. The use of the term mesoangioblasts for these adult cells has led to their unfortunate conflation with the embryonic cells and confusion over their properties. It would be much more accurate and sincere to refer to these cells as pericyte-derived progenitors or pericyte-derived myogenic progenitors.

We understand this Reviewer's point and unfortunately this is a recurrent issue. After more than 50 papers published on the topic, it is now difficult to correct this initial improper definition. Mesoangioblasts are defined as the *in vitro* expanded progeny of cells from the vessel wall. This might be vague as it is, for example, the definition of "mesenchymal stem cells". To better clarify this important point highlighted by the Reviewer, we added the following sentence (Results, page 8): "*Post-natal human mesoangioblasts are considered to be the in vitro expanded progeny of a subset of alkaline phosphatase (ALP)-positive skeletal muscle pericytes (Dellavalle et al, 2007).*"

8. Is the Bmi1 gene used in these studies human or murine? If human, please capitalize to make clear. If murine, please indicate in the results section.

We apologise for not having specified this information in the original manuscript and thank the Reviewer for highlighting it. The Bmi1 cDNA used for this study is murine due to the very high homology and readily availability. Indeed, murine and human Bmi-1 display a high degree of similarity at cDNA (92.4%) and protein level (98%)(Alkema MJ et al., *Hum Mol Genet* 1993; Bhattacharya R et al., *Genes Dis* 2015). We have now specified the murine origin of Bmi1 in the Lentiviral Vectors subsection (page 25) of the Materials and Methods. Please note that this will not impact on the translational potential of our strategy, as the transgene will undergo excision prior to possible transplant and, if needed, could be easily replaced by cloning its human counterpart.

Referee 2

This is a well-written paper exploring solutions for the fact that limited replicative potential in vitro limits our ability to engineer human myogenic cells to be used in therapy. The paper convincingly presents a reversible immortalisation strategy that allows extending the replicative potential of primary myogenic cells. The data is high quality and the experiments well controlled. However there are a few conceptual issues to be addressed: The authors introduce the idea of limited in vitro replicative potential as a key limitation affecting the use of primary myogenic cells, and the bulk of in vitro work is done with satellite cell derived myoblasts. However, the in vivo proof of principle was carried out only with mesangioblasts. Is there a reason for this?

We thank Reviewer 2 for the very positive assessment of our manuscript. We are glad to see that they have rated our paper as “well-written”, the reversible immortalisation strategy as “convincing”, our data as “high quality” and the experiments as “well controlled”. We apologise if this was not highlighted in the manuscript, but actually only initials experiments (e.g. Fig 2, Fig EV1 and Appendix Fig S1) were performed with myoblasts and the bulk of the work was actually then performed with pericyte-derived mesoangioblasts (e.g. Fig 3, 4, 5, 6, Fig EV2, EV3). We started from myoblasts as a proof-of-principle set of experiments given their readily availability (Cudré-Mauroux C et al., *Hum Gen Ther* 2003) and then we moved to pericytes for in vivo experiments. Although we focused on mesoangioblasts for our proof-of-principle in vivo work, we have now also added in vivo data of HAC-corrected DMD myoblasts to strengthen the revised manuscript (Fig 6A).

It appears that extended culture still leads to a significant decrease in myogenic activity even in immortalised cells, suggesting that the maintenance of an undifferentiated status (Pax7 expressing) may not be supported by Bmi1 and TERT expression. This is a significant limitation of the proposed strategy. Is it the same when using mesangioblasts?

We thank Reviewer 2 for raising this interesting discussion point, which we have now addressed in the revised version of our manuscript. We do not think that reversible immortalisation directly interferes with the maintenance of a Pax7-positive undifferentiated status, as Bmi1 has been shown to play a key role in maintaining satellite cell pool in postnatal skeletal muscle (with reduced Pax7+/Myf5- satellite cells found in Bmi1-null mice; Robson LG et al., *Plos One* 2011). What is likely to happen is the immortalisation and subsequent expansion of cells that might have already lost Pax7 expression during the initial expansion prior to lentiviral transduction, hence the following amplification is likely to expand committed progenitors. Recent advances in human satellite cell purification (e.g. Xu X et al., *Stem Cell Reports* 2015) together with immortalisation at very early passages (e.g. P1-2) could improve this outcome. The same process could be less relevant for pericyte-derived mesoangioblasts, whose dependence from Pax7 is less clear (Dellavalle A et al., *Nat Cell Biol* 2007).

Regarding the decrease in myogenic activity, this could also be a consequence of high-density cultures during MMCT (the technique necessary to perform HAC transfer). In the event that this decrease could impact on engraftment and differentiation in vivo, a contingency plan with MyoD-ER-induced rescue of differentiation has been described in the manuscript (Fig 3J and Fig EV2F). We have now discussed the above points in the revised version of our manuscript (page 20-21).

The characterization of the immortalised cells should be extended both during growth and after transgene excision. At a minimum, how many of them are positive for Pax7 MyoD, Myf5 and MyoG? These markers are commonly used to gauge the position of the cells in the myogenic differentiation cascade. What is the spontaneous differentiation rate in culture? Does this change following transgene excision?

We thank Reviewer 2 for having raised this important point. Unfortunately the high-volume of genetic engineering experiments for HAC generation and transfer has partially shifted the focus from important cell biology aspects. We have now added this data in Appendix Figure S1A. As shown in this figure, both before and after Cre-mediated hTERT and Bmi1 transgene excision, DYS-HAC2-genetically corrected immortalised cells are positive for MyoD and Myf5 and negative for Pax7 (which is not surprising given what discussed already in the previous point). Moreover, upon differentiation cells properly express Myogenin and MyHC as markers for terminal skeletal

muscle differentiation, with no differences in their spontaneous differentiation rates. This data is consistent with what already published by Trono and colleagues for the parental, reversibly immortalised DMD myoblasts (Cudré-Mauroux C et al., *Hum Gen Ther* 2003).

Figures 3 g, h strongly suggests that the immortalised cells have a growth advantage. Do cells that retained the immortalizing transgenes following CRE excision have a growth advantage over the successfully excised cells, and will they eventually take over the culture in the absence of counter-selection? Extending the analysis to another time point would reveal this.

We thank Reviewer 2 for raising this interesting point. It is indeed likely that the small percentage of cells that fail to undergo Cre-mediated hTERT and Bmi1 excision might have a growth advantage in the absence of ganciclovir counter-selection. To investigate whether the small percentage of cells that fail to excise the immortalizing cassettes might exhibit a growth advantage in the absence of ganciclovir counter-selection, we studied Bmi1 and hTERT expression levels and cell proliferation at additional time points after IDLV NLS-Cre transduction (4, 5, 6, 7 and 8 weeks). Bmi-1 and hTERT levels were raised from 5-6% of 2 weeks time point (MOI 2.5 IDLV NLS-Cre; Fig 4D, red box) up to 40-50% at 4 weeks time point (MOI 2.5 IDLV NLS-Cre; Appendix Fig S1C) and continued to increase, reaching at 8 weeks the same levels of hTERT and Bmi-1 as immortalised mesoangioblasts not treated with Cre recombinase (Appendix Figure S1C). Proliferation rate of IDLV NLS-Cre mesoangioblasts at 4 weeks from IDLV NLS-Cre transduction was lower than immortalised mesoangioblasts ($42.3 \pm 2.9\%$ vs. $33.5 \pm 1.3\%$; Appendix Fig S1D), consistent with the reduction in hTERT and Bmi1 positive cells. Proliferation rate was restored 8 weeks after IDLV NLS-Cre (IDLV NLS-Cre $33.4 \pm 2.1\%$ vs. non treated $34.4 \pm 1.3\%$; Appendix Fig S1D). These results showed that cells that have retained the immortalising genes have a moderate but significant growth advantage and could eventually take over in the absence of counter-selection. For this reason we have administered ganciclovir within two weeks from Cre-mediated transgene excision (Figure 4D,E).

The engraftment efficiency seems poor when compared with previously published data from this group. Unfortunately a benchmark is not provided (non-expanded cells?). In general, given that the point of this paper is to propose a solid strategy to generate material for cellular therapy, a quantitative assay showing that this strategy is advantageous over existing approaches would be required.

We thank Reviewer 2 for this comment, which gives us the chance to clarify what we believe is the main point of our paper. The aim of our manuscript is not to detail a new pre-clinical cell therapy protocol, but to describe a strategy to overcome the main hurdle limiting HAC transfer into primary human muscle progenitors, leaving to future studies the task of performing detailed functional pre-clinical in vivo experiments based upon this genetic tool. In this context, the cell transplantation experiments were limited to provide proof-of-principle evidence that the product obtained with our approach (i.e. DYS-HAC-corrected DMD myogenic progenitors) were still able to engraft regenerating muscle tissue. Nevertheless, what we presented in Figure 6B is actually in line with the vast majority of published evidence of muscle xenotransplants (about 150 cells / TA central muscle section with a single injection = 5% of myofibres contained human nuclei). We agree with this Reviewer that previous data from our groups have reported better engraftment figures, but those were mostly based upon intraspecific (i.e. mouse cells into mouse muscles; e.g. Tedesco FS et al., *Sci Transl Med* 2011), a procedure more efficient than human xenotransplants in mouse muscles (even in immunodeficient animals, due to their well-functioning innate immunity). Regarding the possibility to benchmark this with existing approaches, we are not sure that this experiment would be informative, as this is not an alternative to muscle stem cell transplant or a better way to do so, but it is rather a strategy to overcome the impossibility to do it after HAC transfer.

3rd Editorial Decision

30 June 2017

Thank you for the submission of your manuscript to EMBO Molecular Medicine and apologies for the delay in providing you with a decision due to the difficulties in obtaining the reviewer evaluations in a timely manner.

We have now received comments from the two Reviewers whom we asked to evaluate your manuscript.

As you will see, in aggregate the two reviewers are now globally satisfied with your extensively revised manuscript. However, both have a few remaining requests aimed at better addressing some of the study limitations. Reviewer 1 in particular, remains critical of the weakness of evidence for in vivo dystrophin expression and suggests a possible experimental avenue to address the issue.

In conclusion, we would be pleased to consider a suitably revised submission that addresses in full the reviewers' requests including with further experimentation, where possible, concerning reviewer 1's specific concern on dystrophin expression.

Provided you deal with the above issues, I am prepared to make an editorial decision on your next final version.

I look forward to reading your revised manuscript as soon as possible.

**** Reviewer's comments ****

Referee #1 (Comments on Novelty/Model System):

One serious issue remains: the in vivo functionality of the HAC with regard to dystrophin expression. This is an issue of technical quality and interpretation. See below.

Referee #1 (Remarks):

This revision includes a major technical innovation, namely the authors have moved all of the components of their conditional immortalization and inducible myogenesis system into the Dystrophin HAC. This significantly elevates the level of innovation and general interest. The authors have also modified or cogently defended in response to several criticisms, including explaining more clearly that differentiation was driven by MyoD in the abstract. However, although technical issues were not the major issue with the previous submission (because novelty, feasibility and overinterpretation were seen as bigger problems), they are now the remaining hurdle. I simply cannot get over the weakness of evidence for which the authors are basing their interpretation of in vivo dystrophin expression (Figure 6).

As indicated previously, it is essential and quite easy to perform single staining with a dystrophin antibody to demonstrate human fiber engraftment. Instead of combining two antibodies in the same fluorescent channel, the authors should immunostain with dystrophin in its channel alone (serial sections if necessary with hLamin A/C, or use a different channel for Lamin A/C), present a representative image, and count fibers and present a plot of fiber number per section.

The current figure shows extremely weak and questionable dystrophin expression, and does not quantify dystrophin + fibers. As additional evidence for dystrophin expression, the authors show a non-quantitative RTPCR. This is not sufficient. Together, it gives the appearance that the HAC is not expressing dystrophin as abundantly as would be expected (perhaps it is getting lost in many cells?) and dystrophin+ fibers were not as abundant as the text suggests, or as the human Lamin A/C+ nuclei counts suggest.

I view this straightforward and quick experiment (dystrophin single channel staining, quantification of dystrophin+ fibers) as an essential remaining revision. Regarding how to respond if the data show that the HAC does not express much dystrophin in vivo, I would not necessarily be opposed to publication, but this limitation in vivo would need to be documented appropriately and discussed.

Referee #2 (Comments on Novelty/Model System):

With the caveat that I mentioned in the comments to author, which is that it is unclear whether this sophisticated cell engineering approach would be viable with primary myogenic cells. On the other hand, the generation of myogenic cells from hES cells is at hand and that would bypass this limitation.

Referee #2 (Remarks):

In their response, the authors changed the focus of the paper from the implementation of an efficient therapy to a proof of principle for a new approach to cell engineering with feature-laden artificial chromosomes. In doing so, they have added a significant amount of data that is not merely an enhancement of the previous version of the paper but it also adds a new conceptual dimension. In a way, this clarifies why the efforts to fully exploit the peculiar characteristics of mesangioblasts were limited in the original submission. I found the new data to be of high quality and convincing, making this version of the paper worthy of publication. My only question is whether such an approach would be possible with, for example, primary myoblasts which cannot be expanded as much as mesangioblasts and are therefore unlikely to perform following the required amount of *in vitro* selection. This should be discussed.

Additional correspondence (author)

14 September 2017

Please accept our apologies for the delay. We experienced a ridiculous number of technical issues, from anti-dystrophin antibody not being delivered for several weeks, to culture media (MegaCell) production and shipment delayed by Sigma Aldrich.

Nonetheless, we managed to set up new *in vivo* experiment requiring sourcing of expensive (and not readily available) NSG immunodeficient mice. We have now started the processing and analysis of the samples of this *in vivo* experiment, with promising preliminary results. I will not be in the lab for a week, but we will be ready to submit this newly revised version of the manuscript by the end of this month.

Hope this timeline is still ok with you?

Apologies again for the delay and thanks in advance for your patience.

Additional correspondence (author)

28 September 2017

Following our recent correspondence below, I would like to give you further updates on the status of our revision.

As you may remember, the outstanding issue was Reviewer 1's request to see more evidence of dystrophin expression *in vivo*. To address this query we eventually setup new *in vivo* experiments in immunodeficient mice. As mentioned in my previous email, this also triggered a number of unexpected technical and logistical issues, with major delays in receiving mice, culture media and antibodies (last two still not received, but we were luckily helped by colleagues). Intramuscular injection of HAC-corrected cells resulted in a good number of engrafted cells (similarly to what we reported already), although the majority of them did not fuse with host myofibres - hence we could not properly assess dystrophin production *in vivo*. However, we did perform another (possibly more stringent) *in vivo* experiment, injecting HAC-corrected cells in subcutaneous matrigel plugs in immunodeficient mice (as recently reported by Sacchetti B et al., *Stem Cell Reports* 2016). Although heterotopic, this assay is actually more specific than a simple intramuscular injection, as it tests myogenic potential and dystrophin production of the cells of interest in a cell-autonomous fashion (i.e. if you see dystrophin it can only be produced by donor cells, as there are no host myofibres providing background noise). This experiment was successful and we have seen human lamin A/C & dystrophin double-positive myotubes in all injected mice. Please find enclosed a new draft figure and legend summarising the results, which we will incorporate in Figure 6 - we believe that this addresses the Reviewer's concern.

However, while analysing this experiment we also realised that there is a remote possibility that some residual dystrophin could be produced by the native DMD locus. Those cells have a deletion from exon 5 to 7 and there was an old paper reporting some residual expression of dystrophin in patients with deletions in exons 3-7, possibly caused by an ATG in exon 8 (Winyard AV et al., *Am J Hum Genet* 1995). Although we did not observe any dystrophin expression in previous *in vitro* experiments (Figure 2F) and this is in keeping with what the group of Dider Trono (who first

characterised that line) originally reported (Cudré-Maroux C et al., Hum Gen Ther 2003), we believe that it would be a pity not to assess this properly with an ad-hoc experiment. Therefore, we would be extremely grateful if you could please give us a couple of weeks to assess this eventuality and, if needed, run additional controls to test HAC-specific dystrophin expression using antibodies and primers specific for the region deleted in the parental population. Even though this was not a request of the Reviewer, we think that this extra thoroughness will be in the best interest of both us and your Journal.

We are aware that you will be leaving your editorial role at EMM soon and understand that it would have been ideal to reach a final decision before that date, so please accept our apologies for any inconvenience or extra work that such an option might entail.

We thank you very much for your consideration.

2nd Revision - authors' response

07 November 2017

Following our recent correspondence, please find enclosed our revised manuscript EMM-2016-07284-V4 by Sara Benedetti et al. Let me thank you one more time for the additional time you and Roberto have given us to address the final Reviewers' requests. Once again, we were extremely pleased to see the very positive assessment of our revised manuscript, which we think is now further improved.

You might remember that the main outstanding issue was a request from Reviewer 1 to have more evidence of dystrophin expression *in vivo*, specifically to see co-localisation of lamin A/C+ nuclei with dystrophin+ fibres. To address this query we setup new *in vivo* experiments in immunodeficient mice. As mentioned in my previous email, this also triggered a number of unexpected technical and logistical issues, with major delays in receiving mice, culture media and antibodies. Intramuscular injection of HAC-corrected cells resulted in a good number of engrafted cells (similarly to what we reported already), although the majority of them did not fuse with host myofibres - hence we could not properly assess dystrophin production *in vivo* with that specific assay. Nonetheless, we performed another (possibly more stringent) *in vivo* experiment, injecting HAC-corrected cells in subcutaneous matrigel plugs in immunodeficient mice (as recently reported by Sacchetti B et al., Stem Cell Reports 2016). This heterotopic assay is actually more specific than a simple intramuscular injection, as it tests myogenic potential and dystrophin production of the cells in a cell-autonomous fashion (i.e. dystrophin can only be produced by donor cells, as there are no host myofibres providing background noise). This experiment was successful and we have seen human lamin A/C & dystrophin double-positive myotubes in all injected mice (now in Fig.s 6E and S3). Moreover, we have also performed an experiment to demonstrate that the observed dystrophin is indeed produced by the HAC and not by events of spontaneous exon-skipping which might theoretically restore the reading frame in the donor with mutation in exons 5-7. Importantly, this dystrophin transcript analysis confirmed the presence of an out-of-frame mutation and ruled out the possibility of restoration of the reading frame by skipping of exon 8 (now in Fig EV1E), in keeping with: 1) our previous observation (Figure 2F); 2) what the group of Dider Trono (who generated and characterised that line) originally reported (Cudré-Maroux C et al., Hum Gen Ther 2003); 3) the actual clinical phenotype and biochemical readouts reported in patients (Muntoni et al., J Med Genetics 1994).

Reviewer 2 gave us an excellent assessment and only requested us to discuss if our next-generation HAC (i.e. Fig 7) would be applicable to primary myoblasts as opposed to mesoangioblasts, which we have done on page 21 of the Discussion section.

Other minor changes (highlighted in blue font) in this version of the manuscript include:

- Correction of typos and re-wording of some sentences in the manuscript;
- Improved consistency with labelling in figures and correction of minor inaccuracies;
- Six new references (and one removed);
- New PCRs in Fig EV3A, confirming absence of contaminating donor CHO cells in the clones;
- Western blot in current Fig 2E has been updated with a new panel showing additional controls and higher resolution.

Corresponding Author Name: Francesco Saverio Tedesco and Giulio Cossu

Journal Submitted to: EMBO MOLECULAR MEDICINE

Manuscript Number: EMM-2016-07284-V2-Q